

# Non-linear Climatic Response to the Weakening of the Atlantic Meridional Overturning Circulation During Glacial Times

Yanxuan Du[1,2], Josephine R. Brown[1,2], Laurie Menviel[3,4], Himadri Saini[1,2,3,4], Russell N. Drysdale[1], David K. Hutchinson[3], and Calla N. Gould-Whaley[1,5]

[1]School of Geography, Earth and Atmospheric Sciences, University of Melbourne, Melbourne, Victoria, Australia
[2]Australian Research Council Centre of Excellence for Weather of the 21st Century, University of Melbourne, Melbourne, Victoria, Australia
[3]Climate Change Research Centre, University of New South Wales, Sydney, New South Wales, Australia
[4]The Australian Centre for Excellence in Antarctic Science, University of Tasmania, Hobart, Tasmania 7001, Australia
[5]School of Biological, Earth and Environmental Sciences, University of New South Wales, Sydney, New South Wales, Australia

*Correspondence to*: Yanxuan Du (yanxuand@student.unimelb.edu.au)

**Abstract.** The climatic response to the weakening of the Atlantic Meridional Overturning Circulation (AMOC) is investigated under glacial conditions representative of Heinrich Stadial 5 using a fully coupled Earth System Model (ACCESS-ESM1.5), with a focus on Southern Hemisphere and Australian hydroclimate. We find that the climatic response to an AMOC slowdown or shutdown, respectively representing Dansgaard-Oeschger (D-O) and Heinrich stadials, is non-linear. Global mean temperature and precipitation anomalies increase linearly with AMOC weakening; however, crossing the threshold of AMOC shutdown results in non-linear and more complex atmospheric circulation and climate responses. A shutdown of the AMOC in the simulations leads to an enhanced and expanded northern winter Hadley Cell (HC), with a southward shift of its ascending branch. The southern winter HC is weaker but increased in width due to a northward shift of the ascending branch due to AMOC shutdown. This change in the HC drives seasonal variations in the Northern and Southern Hemispheres subtropical high pressure systems and subsequently, changes in the cross-equatorial atmospheric circulation, as well as the Southern Hemisphere mid-latitude westerly winds and other climate features such the monsoon systems. The simulation results are broadly consistent with available proxy records for Heinrich and D-O stadials as well as previous model simulations. The study demonstrates the potential location of a threshold in the climate system between linear weakening and nonlinear shutdown of AMOC with differing climate impacts, further highlighting the importance of not crossing the threshold of AMOC shutdown in the future.

## 1 Introduction

The Last Glacial Period (LGP, ~115,000 to 11,700 years ago) was characterised by repeated millennial-scale climate oscillations, which are commonly referred to as Dansgaard-Oeschger (D-O) variability (Dansgaard et al., 1993). During each D-O event, Greenland temperature experienced an abrupt warming (a Greenland Interstadial), the temperature then gradually decreased within a few centuries before it ended in an abrupt cooling to near-full glacial (Greenland stadial) conditions, accompanied by a weakening of the Atlantic Meridional Overturning Circulation (AMOC) (Barker et al., 2009; Corrick et





al., 2020; Menviel et al., 2014b). During Greenland cooling events (stadials), reduced northward oceanic heat transport induced by a weaker AMOC resulted in heat accumulation in the South Atlantic, thus leading to Antarctic warming with a lag of ~210 years (Barker et al., 2009; EPICA Community Members, 2006; Stocker and Johnsen, 2003; WAIS Divide Project Members, 2015). During some of the coldest D-O stadials, massive iceberg discharges into the North Atlantic Ocean led to a substantial weakening or shutdown of the AMOC (Barker et al., 2015; Broecker et al., 1992; Broecker, 1994; Henry et al., 2016; Zhou and McManus, 2024). These episodes are known as Heinrich events (HEs; Bond et al., 1992; Heinrich, 1988), and the cold time periods that contains the HEs are named Heinrich stadials (HSs) (Heinrich, 1988). Despite some debates about the Heinrich terminologies in literatures (e.g. Andrews and Voelker, 2018; Hodell et al., 2017), we follow the consistent usage of these terms in previous studies (Barker et al., 2015; Sanchez Goñi and Harrison, 2010) throughout the paper as described above.

During D-O stadials, paleoclimate proxy records suggest cooler and drier conditions in the North Atlantic and many regions over Eurasia and North America (Izumi et al., 2023; Martrat et al., 2007). Temperature changes in Greenland ice core records show relatively similar amplitudes between D-O and HSs (Kindler et al., 2014; Svensson et al., 2008). However, sea surface temperature (SST) estimates from marine sediment records from the Iberian Margin display a temperature decrease up to twice as large in HSs compared to D-O stadials (Martrat et al., 2007). Climatic response during HSs is similar in direction but greater in magnitude than D-O stadials due to the large magnitudes of change in AMOC strength, and thus heat redistribution via changes in oceanic and atmospheric circulation (Izumi et al., 2023; Menviel et al., 2020; Pedro et al., 2018; Stocker and Johnsen, 2003). Whilst a seasonal southward migration of the Intertropical Convergence Zone (ITCZ) has been inferred during mild stadial conditions, year-round displacement of the ITCZ position is suggested during HSs (Deplazes et al., 2013). Nevertheless, it remains less certain whether the climate response to Heinrich and D-O stadials is linearly related to AMOC weakening.

Heinrich stadial 5 occurred at a time (~48.8-47.6 ka; Menviel et al., 2014b; Sanchez Goñi and Harrison, 2010) during Marine Isotope Stage 3 (MIS3, ~59.4-27.8 ka), received higher summer insolation in both hemispheres relative to pre-industrial (PI) levels due to greater obliquity (Berger, 1978). The atmospheric $CO_2$ concentration was lower (Köhler et al., 2017). Global sea level was ~60-65 m lower than PI (Shakun et al., 2015), with extensive Laurentide and Scandinavian ice sheets (Gowan et al., 2021). The global temperature is 2.7 °C cooler as a result of changes in all the boundary conditions (orbital, ice sheet albedo, etc.) in model simulations of 49 ka (Saini et al., 2025b). Paleoclimate records at Girraween Lagoon in northern Australia suggest relatively drier background climates during MIS3 relative to PI, with wetter conditions during HSs than interstadials during MIS3 (Bird, 2025). This is consistent with new speleothem records from southern Australia, suggesting wetter conditions during HS5 (Gould-Whaley et al., 2025, in prep.).

Many modelling studies have investigated the climatic response to AMOC variability in the past (Cheng et al., 2007; Chiang et al., 2014; Chiang and Friedman, 2012; Du et al., 2025; Kageyama et al., 2013; Lee et al., 2011; Menviel et al., 2014b; Saini et al., 2025a), including during the pre-industrial period (Ben-Yami et al., 2024; Jackson et al., 2015; Orihuela-Pinto et al., 2022; Saini et al., 2025a). A weakened AMOC leads to cooling in the North Atlantic with slow



warming in the South Atlantic due to reduced northward oceanic heat transport (Stocker and Johnsen, 2003). The ITCZ shifts southwards towards the warmer hemisphere due to changes in the interhemispheric temperature gradient, leading to an increase in Southern Hemisphere (SH) monsoon precipitation and reduced precipitation in Northern Hemisphere (NH) monsoons (Cheng et al., 2012; Zhang et al., 2023). This is consistent with paleoclimate records, which suggest weakened Indian, East Asian, and West African monsoons (Pausata et al., 2011; Xue et al., 2025; Zhao et al., 2010), and increased Indo-Australian monsoons (Denniston et al., 2013, 2017; Treble et al., 2017) during D-O and HSs.

The ITCZ is the region where low-level trade winds converge and form ascending branches of the Hadley Cell (HC) (Lionello et al., 2024). A southward shift in the ITCZ also moves the rising branch of the HC southwards, leading to changes in the strength of the northern and southern HCs (Chiang et al., 2014; Chiang and Friedman, 2012; Lee et al., 2011; Saini et al., 2025a; Zhang and Delworth, 2005; Zhang et al., 2023). This anomalous cross-equatorial atmospheric heat transport partially compensates for the reduced northward oceanic heat transport (Lee et al., 2011). Changes in the HC also greatly influence large-scale climate features such as the subtropical high pressure systems (subtropical ridge; STR) and atmospheric circulation, including the mid-latitude westerly winds, affecting global hydroclimate accordingly (Chiang et al., 2014; Chiang and Friedman, 2012; Lionello et al., 2024; McGee et al., 2014; Orihuela-Pinto et al., 2022; Zhang and Delworth, 2005). A southward shift and strengthening of the SH westerly winds during austral winter (June-July-August; JJA) are suggested by limited proxy and modelling studies during North Atlantic cold events (Anderson et al., 2009; Lee et al., 2011; Timmermann et al., 2007; Whittaker et al., 2011). A consistent response of the SH westerlies to an AMOC slowdown is still under investigation in different modelling studies (e.g. Lee et al., 2011). Moreover, there is a need to improve our understanding of climatic processes in the Indo-Pacific region from model simulations (Ben-Yami et al., 2024; Izumi et al., 2023; Jackson et al., 2015; Kageyama et al., 2013; Zhang and Delworth, 2005).

In this study, we investigate the climatic changes associated with D-O and HSs at 49 ka - coinciding with Heinrich stadial 5 - using the Earth System Model ACCESS-ESM1.5. This model has demonstrated high skill in simulating the SH and Australian climates (Grose et al., 2020; Ziehn et al., 2020). We compare the linearity in the climatic responses to changes in AMOC transport, taking an AMOC shutdown as an analogue of HS conditions and an AMOC weakening as an analogue of a D-O stadial. The results expand on previous studies by focusing on the SH climate response beyond the Atlantic Ocean, including exploring the implications of large-scale climate changes to Australasian climates. The results aim to provide a detailed description of SH climate response associated with AMOC variability under glacial climates from a modelling perspective and will be briefly compared with proxy records to assess model-data agreement.

## 2 Methods

### 2.1 Model description

In this study, we use the Australian Community Climate and Earth System Simulator (ACCESS) Earth System Model (ESM) version 1.5, which participated in the Coupled Model Intercomparison Project Phase 6 (CMIP6) and includes an interactive





carbon cycle (Ziehn et al., 2020). The resolution of the atmospheric model component (UK Met Office Unified Model, UM) is 1.875° longitude × 1.25° latitude × 38 levels (Martin et al., 2010; The HadGEM2 Development Team: G. M. Martin et al., 2011), coupled to the land surface model, which is the Community Atmosphere Biosphere Land Exchange (CABLE) model (Kowalczyk et al., 2013). The ocean component uses the Geophysical Fluid Dynamics Laboratory Modular Ocean Model 5.1 (MOM5.1; Griffies, 2012) and sea ice component uses the Los Alamos Community Ice Code model CICE4.1 (Hunke and Lipscomb, 2010). The ocean and sea-ice components use a common horizontal grid of 360 × 300 cells at a nominal 1° resolution, with meridional refinement of grid spacing down to 0.33° at the equator and down to ~0.4° in the Southern Ocean and a tripolar grid with land-based poles at 65° N over Eurasia and North America (Ziehn et al, 2020). More details of the model description and performance are provided in Ziehn et al. (2020). Previous studies have performed freshwater hosing experiments to alter the AMOC strength in pre-industrial, interglacial boundary conditions, historical and future simulations using this model (Du et al., 2025; Pontes and Menviel, 2024; Saini et al., 2025a), with acceptable AMOC sensitivity to freshwater perturbations within the range of CMIP6 models (Weijer et al., 2020).

## 2.2 Heinrich 5 climate simulation at 49 ka

The model was first integrated under 49ka boundary conditions as described in detail in Saini et al., (2025b). More specifically, the model is forced by 49ka orbital parameters, greenhouse gases concentration, ice-sheet extent and topography, vegetation, appropriate land-sea mask and river runoff (Table 1).

**Table 1.** Full boundary conditions for the 49ka_control experiment relative to pre-industrial simulation (PI).

|  | 49 ka | PI |
|---|---|---|
| **Orbital parameters** |  |  |
| Eccentricity | 0.01292 | 0.01674 |
| Obliquity (°) | 24.435 | 23.459 |
| Perihelion – 180 (°) | 62.451 | 100.33 |
| **Greenhouse gases** |  |  |
| $CO_2$ | 199 | 284.3 |
| $N_2O$ | 237 | 273 |
| $CH_4$ | 432 | 284.3 |
| **Albedo and vegetation** | 49 ka | PI |
| **Salinity** | 49 ka | PI |
| **Topography and runoff** | 49 ka | PI |





The 49ka simulation was run for a total of 1555 years with step-wise changes in the boundary conditions (Saini et al., 2025b). The 49ka_control simulation with full boundary conditions (49ka-full in Saini et al., 2025b) was run for 760 model years, with the last 100 years showing relatively stable changes in surface air temperature and sea-surface temperature changes (Saini et al., 2025b, Fig. A1) which we consider as a quasi-equilibrium state for 49 ka climate. This 49ka_control displays a stronger AMOC strength than PI (31 Sv compared to 21 Sv for PI), which is considered to represent interstadial conditions at 49 ka. The stronger AMOC at 49 ka is due to changes in the topography and expansion of the ice sheets in the model, which enhances the North Atlantic gyres and cyclonic circulation in the Labrador Sea (Saini et al., 2025b). Simulated globally averaged surface air temperatures are 2.7 °C lower than pre-industrial, and the SST is 1.2 °C colder. Drier conditions are simulated over the NH high latitudes and most land areas of the SH relative to PI (Saini et al., 2025b, Fig. 2c).

## 2.3 Experimental design for D-O and HSs at 49 ka

For this study, North Atlantic meltwater hosing experiments (Table 2) are performed following attainment of model quasi-equilibrium state conditions (49ka_control). Freshwater fluxes were added into the North Atlantic (50° N-70° N, 70° W-0° W) to alter the AMOC strength in the model, similar to previous studies (Du et al., 2025; Pontes and Menviel, 2024; Saini et al., 2025a).

Previous modelling studies suggest that D-O stadials are associated with a ~50 % AMOC slowdown, while HSs are more likely to close to a complete shutdown of the AMOC (e.g. Malmierca-Vallet et al., 2023; Menviel et al., 2020), even though the exact changes in the AMOC transport during D-O and HSs are still poorly constrained from proxy records (Henry et al., 2016; Lynch-Stieglitz, 2017).

To simulate D-O stadials, we add 0.2 Sv and 0.3 Sv of freshwater in the northern North Atlantic under constant 49ka boundary conditions (49ka_0.2Sv and 49ka_0.3Sv experiments). This leads to ~32 % and ~50 % AMOC reductions relative to the 49ka_control, respectively (see detailed changes in Table 2). We also perform an AMOC shutdown experiment with a constant 0.4 Sv freshwater forcing to simulate HSs (49ka_shutdown; Table 2). See Fig. S1 for changes in the mixed layer depth in each simulation.

**Table 2**: North Atlantic freshwater hosing experiments performed at 49 ka. The forcing strength (Sv), release duration (yr), analysis time period, and the AMOC mean strength (Sv) at analysis period (change relative to 49ka_control) are listed for each experiment.

| Experiments | North Atlantic meltwater input (Sv) | Freshwater release duration (yr) | Analysis time period | AMOC strength (change relative to control) (Sv) |
|---|---|---|---|---|
| 49ka_0.2Sv | 0.2 | 200 | yr 150-200 | 21 (-10) |



| 49ka_0.3Sv | 0.25 | 100 | yr 450-500 | 17 (-14) |
|------------|------|-----|------------|----------|
|            | 0.3  | 200 |            |          |
| 49ka_shutdown | 0.4 | 500 | yr 450-500 | 2 (-29) |

The AMOC evolution for each simulation is shown in Fig. 1. In 49ka_shutdown, the AMOC takes ~100 model years to shut down (below 5 Sv was considered shutdown, Saini et al., 2025a). The simulation is then continued for another 400 model years for the model to fully respond to the shutdown, especially at southern high latitudes. The last 50 years of the simulation was used for analysis (dark blue in Fig. 1).

To explore pseudo D-O stadial conditions, we consider both the 49ka_0.2Sv, and 49ka_0.3Sv experiments. For 49ka_0.2Sv, we analyse years 150-200 (dark red in Fig. 1). To further weaken the AMOC, 49ka_0.2Sv is continued first forced with 0.25 Sv meltwater input, then with 0.3 Sv (see Fig. 1). The analysis period for 49ka_0.3Sv is the 50-yr interval year 450-500 (dark brown in Fig. 1), which corresponds to the same integration time as the 49ka_shutdown experiment to allow for better comparisons.

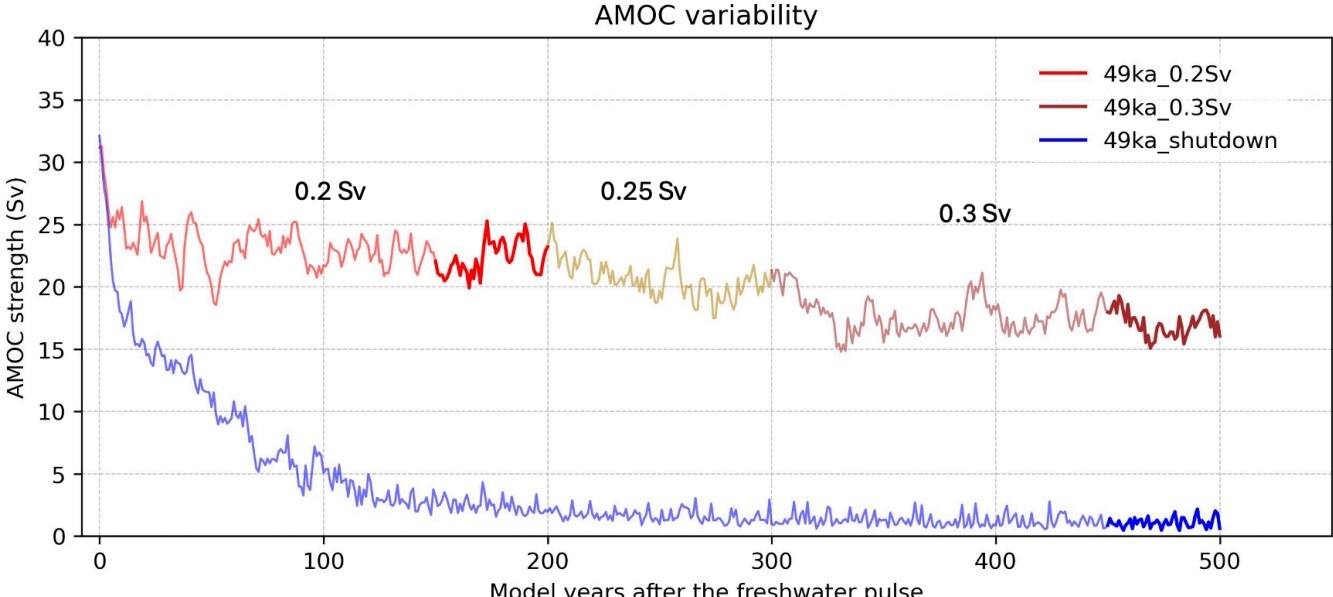

**Figure 1:** Time series of AMOC strength (Sv) after the freshwater pulse at year 0 for the D-O stadial (red-yellow-brown) and HS (blue) experiments. Darker colours indicate 50-year analysis period shown in Table 2. AMOC strength is calculated as the maximum value of the overturning streamfunction between 30° N and 50° N and below 500 m depth, in the Atlantic basin.



**2.4 Calculation metrics**

The NH and SH HC strength is calculated as the maximum and minimum value of the atmospheric mass streamfunction at
500 hPa in DJF and JJA, respectively. The width of the HC is calculated from the latitudes of zero mass streamfunction at
mid-tropospheric level (500 hPa) as edges of the ascending and descending branches (Frierson et al., 2007; Seo et al., 2014).
The mass streamfunction is interpolated to 0.5º latitude resolution for the calculations.

SH westerlies strength is calculated as the 850 hPa zonal mean maximum wind speed between 40º S to 60º S, whilst
the position of the SH westerly winds refers to the latitude of the maximum wind speed interpolated to 0.5º latitude
resolution, consistent with previous studies (e.g. Du et al., 2024; Saini et al., 2025b). Changes in the SH westerlies are
calculated across different ocean basins, the longitude range for the Atlantic Ocean covers between 70º W and 0º, the Pacific
Ocean 130º E-70º W, and the Indian Ocean from 0 to 130º E.

The STR position and intensity over Australia is calculated as the maximum mean sea level pressure (MSLP) value
zonally averaged over 140º E-150º E between 10º S and 45º S. The zonal mean MSLP values are then interpolated to 0.5º
latitude resolution, followed by a fitted cubic spline to detect the latitude of maximum pressure. The latitude is the STR
location, and the pressure value is the STR intensity (Grose et al., 2015).

Finally, monsoon domains are defined as regions in which the wet monsoon season precipitation rate exceeds the
dry season rate by at least 2.5 mm/day, and is responsible for at least 55 % of the annual precipitation (Wang et al., 2011;
Yeung et al., 2021).

**3 Results**

**3.1 Overview of the climate response during pseudo D-O and HSs**

First, we provide an overview of annual temperature and precipitation changes in our D-O and HSs simulations relative to
the 49ka_control climate and compare the linearity between the climatic responses to each hosing experiment by normalising
the temperature and precipitation response to a "per Sv" AMOC decrease. Then, we evaluate the changes in seasonal large-
scale atmospheric circulations by investigating the changes in Hadley Cell, MSLP, and SH westerly winds to AMOC
weakening. Finally, we investigate the influence of changes in these climate features on Australasian climates.

The meridional oceanic heat transport to the North Atlantic is reduced by ~77 % (~1 PW at 30° N) in the HS
(49ka_shutdown) experiment associated with AMOC shutdown (see Fig. S2). This change is larger than previous model
simulations with AMOC shutdown that found a ~40 % (~0.8 PW) reduction in meridional heat transport (Kageyama et al.,
2013; Menviel et al., 2008, 2020). Reduced heat transport leads to a widespread significant NH mean cooling of 3.8 °C, with
maximum annual mean surface air temperature (SAT) decrease of up to 26.3 °C in the North Atlantic (Fig. 2a), while the SH
display a significant warming (mean 0.43°C) from 0 to 55°S. Simulated SST show up to 15.4 °C cooling in the North



Atlantic, and up to 6.5 °C warming in the South Atlantic (Fig. S3a). Over Antarctica, significantly cooler conditions are simulated in the Ross and Weddell Seas (Fig. 2a).

195       The temperature patterns in the two AMOC slowdown simulations (representative of D-O stadials) show generally similar responses to the interhemispheric response in HS, with NH cooling and some SH warming anomalies relative to 49ka_control (Fig. 2b, 2c). The AMOC reduction in D-O stadials seems to be too small to only induce some significant warming in the Indian and eastern Pacific beyond the Atlantic Ocean, with no significant annual mean Antarctic warming is simulated (Fig. 2b, 2c). This is expected, as ~50 % of AMOC slowdown in 49ka_0.3Sv experiment only induces 25 %

reductions in the northward oceanic heat transport at 30° N (see changes in meridional oceanic heat transport in Fig. S2), which is insufficient to lead to the SAT warming over Antarctica observed in the shutdown simulation (Fig. 2a).

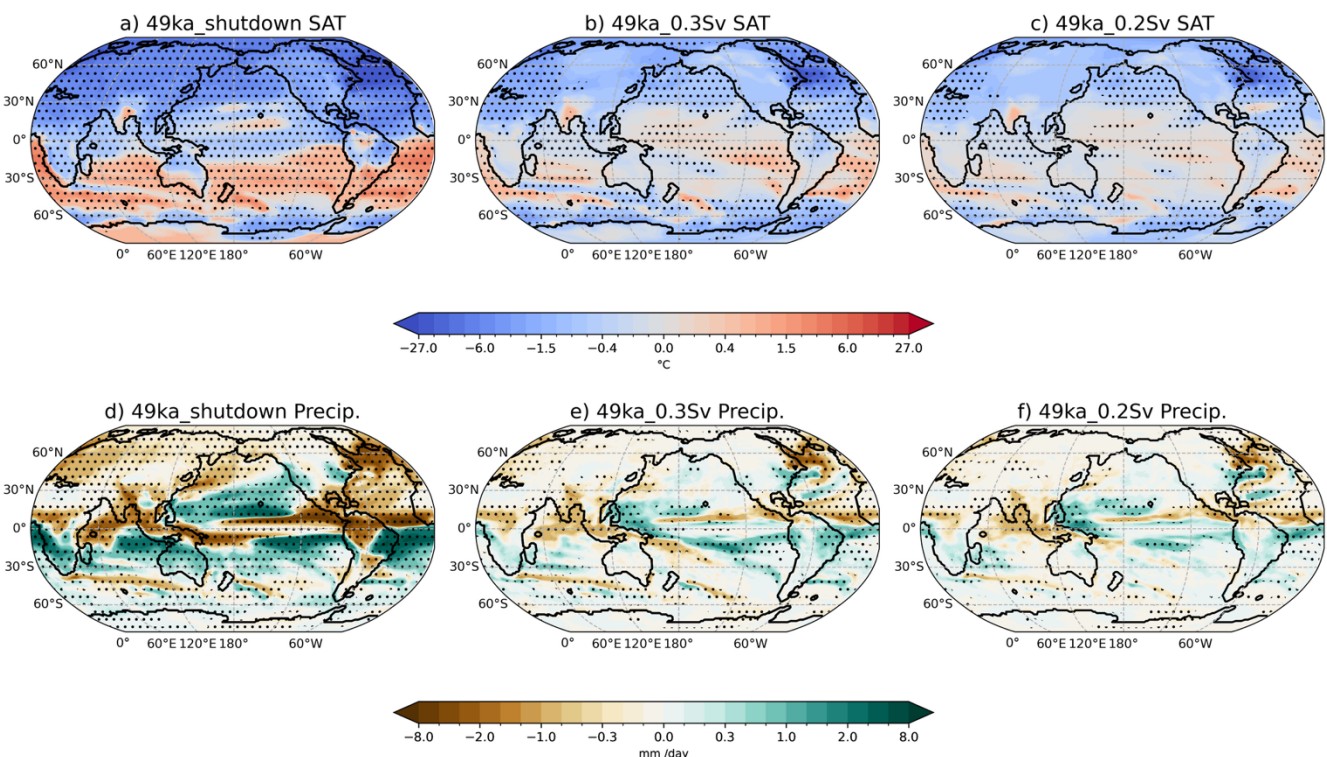

**Figure 2:** Annual (a)-(c) Surface Air Temperature (SAT; in ºC) and (d)-(f) precipitation (in mm/day) anomalies relative to 49ka_control in each simulation. Stippling indicates statically significant differences from the control at the 95 % confidence level according to the Student's t-test.

      Precipitation patterns are more diverse between each simulation (Fig. 2d-2f). In response to an AMOC shutdown

(Fig. 2d), a significant precipitation reduction is simulated to the north of the equator while the precipitation is greatly enhanced to the south. This indicates a southward shift of the mean ITCZ position, extending from the Atlantic to every



ocean basin. This is due to the large interhemispheric temperature gradient anomalies, which push the ITCZ towards the warmer hemisphere. Widespread drying is simulated over much of the NH associated with the cooling, which is consistent with previous modelling results with a weaker AMOC (e.g. Jackson et al., 2015).

In the D-O stadial simulations with ~50 % and 32 % AMOC reductions (Fig. 2e, 2f), the southward displacement of the ITCZ is only evident over the Atlantic Ocean, extending into eastern equatorial Pacific. The precipitation patterns show similar responses, with slightly reduced precipitation in the South Pacific Convergence Zone (SPCZ) region (10° S-30° S, 155° E-140° W) in the slowdown simulations (Fig. 2e, 2f), which is not evident in the shutdown experiment (Fig. 2d). The precipitation rate is significantly increased in the tropical western Pacific in the slowdown simulations (Fig. 2e, 2f),
corresponding to the warming anomalies (Fig. 2b, 2c).

     To further quantify the linearity of changes, we normalise the temperature and precipitation response to a "per Sv" AMOC decrease relative to 49ka_control (Fig. 3). The D-O stadial simulations with 10 Sv and 14 Sv decrease in AMOC strength show the same degree of annual mean temperature and precipitation changes per Sv change (details in Table S1), which indicate linear relationships. This can be seen from the regions of significant changes in the 49ka_0.2Sv experiment
(Fig. 2c, 2f) being enhanced in magnitude and expanded in areal extent in the 49ka_0.3Sv simulation (Fig. 2b, 2e), and similar normalised spatial patterns between the two experiments (Fig. 3b, 3c, and 3e, 3f). Nevertheless, crossing the tipping point of a full shutdown of the AMOC under glacial conditions leads to around 1.3 times the global annual average temperature cooling per Sv AMOC decrease compared to D-O experiments (see Table S1). More specifically, NH temperature is simulated to show up to 1.6 times more cooling in 49ka_shutdown, while SH (from 0 to 55° S) experiences up
to 6.8 times warmer temperatures per Sv than the slowdown experiments (see Fig. 3a-c for normalised spatial patterns). This large difference in SH temperature changes between the D-O and HSs simulations are attributed by a simulated SH mean cooling in D-O experiments while SH warming in the HS (Table S1; Fig. 3a-c). The globally averaged precipitation in AMOC shutdown is also ~1.3 times the reductions in AMOC slowdown simulation per Sv of AMOC decrease, with ~2 times drier NH while 3 times wetter SH per Sv compared to the D-O slowdown simulations (Table S1; Fig. 3d-f).




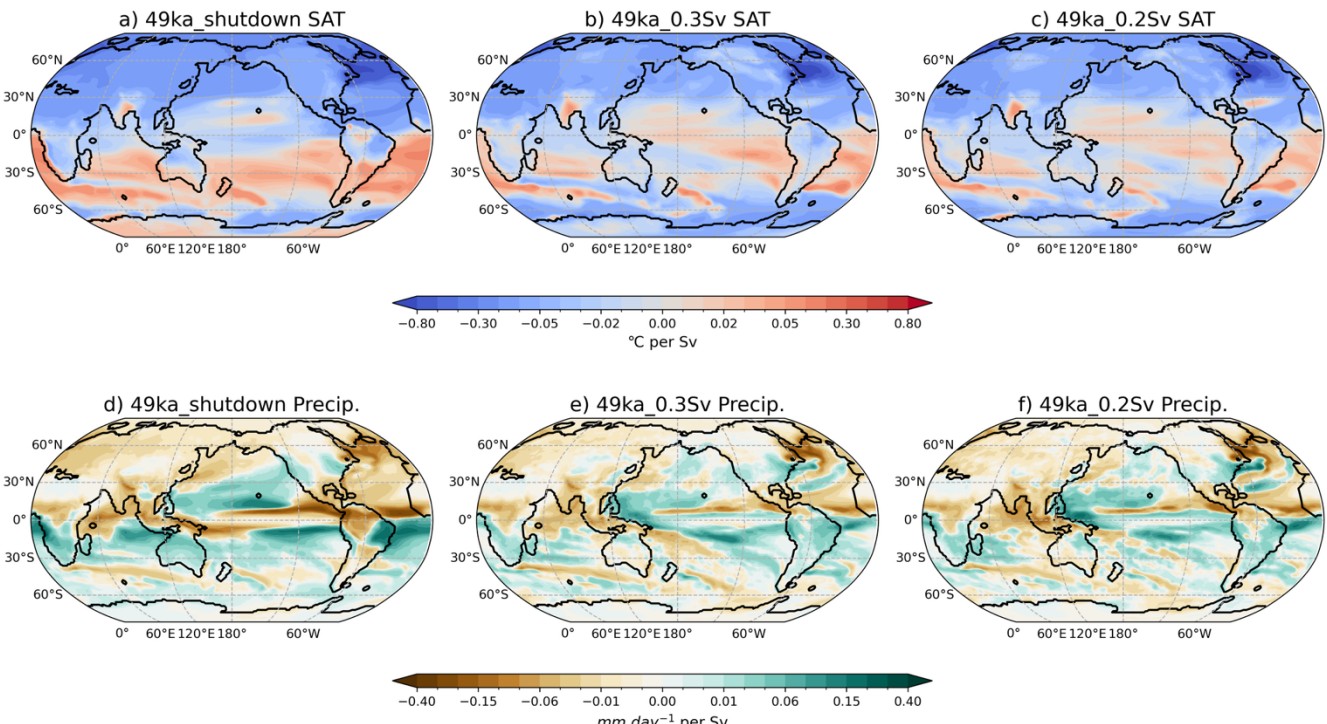

**Figure 3:** Normalised annual surface air temperature anomalies (SAT; ºC per Sv), and precipitation anomalies per Sv AMOC decrease (mm day⁻¹ per Sv) relative to 49ka_control in each simulation.

## 3.2 Non-linear response in large-scale seasonal atmospheric circulations

In this section, we investigate the drivers of the diverse hydroclimate response in the simulations by examining the changes in the HC, and the associated changes in the MSLP, 850 hPa wind circulations, including the SH mid-latitude westerly winds, and the STR using the metrics described in Section 2.3.

### 3.2.1 Alterations in the Hadley Cell

In response to AMOC weakening, the NH winter (DJF) HC strengthens while the SH winter JJA HC weakens in all simulations (Fig. 4 & Fig. 5). The shift in the ITCZ is only evident in the Atlantic basin, extending into the eastern equatorial Pacific in the slowdown simulations from the annual mean precipitation spatial patterns (Fig. 2d-2f). As the ascending branch of the HC, the ITCZ at 850 hPa in DJF season shifts southwards by 0.82° and 0.34°, respectively in the 49ka_0.2Sv and 49ka_0.3Sv slowdown simulations, compared to a 3.57° southward shift to ~15.8° S in the 49ka_shutdown simulation (thick lines in Fig. 4a, 4c, 4e). The global mean position of the JJA ITCZ shifts northwards by 0.97° to around 20.14° N in



49ka_shutdown experiment, while a slight southward displacement of 0.38° is simulated in 49ka_0.3Sv and 0.05° northward shift in 49ka_0.2Sv simulation at 850 hPa (thick lines in Fig. 4b, 4d, 4f).

In the boreal winter (DJF) season, the northern HC strengthens and expands in width at 500 hPa due to AMOC weakening (Fig. 5). The DJF northern HC central position with maximum HC strength shifts southwards by 10° to 1.88° S in the 49ka_shutdown simulation, however, remains unchanged at around 8.13° N in the slowdown simulations (Fig. 4a, 4c, and 4e). The northern HC strength increases by 20 % due to AMOC shutdown compared to an increase of less than 2.3 % in the slowdown simulations (Fig. 5). The NH HC width increases slightly between the two slowdown simulations but is more than doubled in the shutdown simulation (Fig. 5), and is dominated by the southward shift in the ascending branch of the NH HC at 500 hPa, albeit with little change in location of the descending branch (details in Table S2).

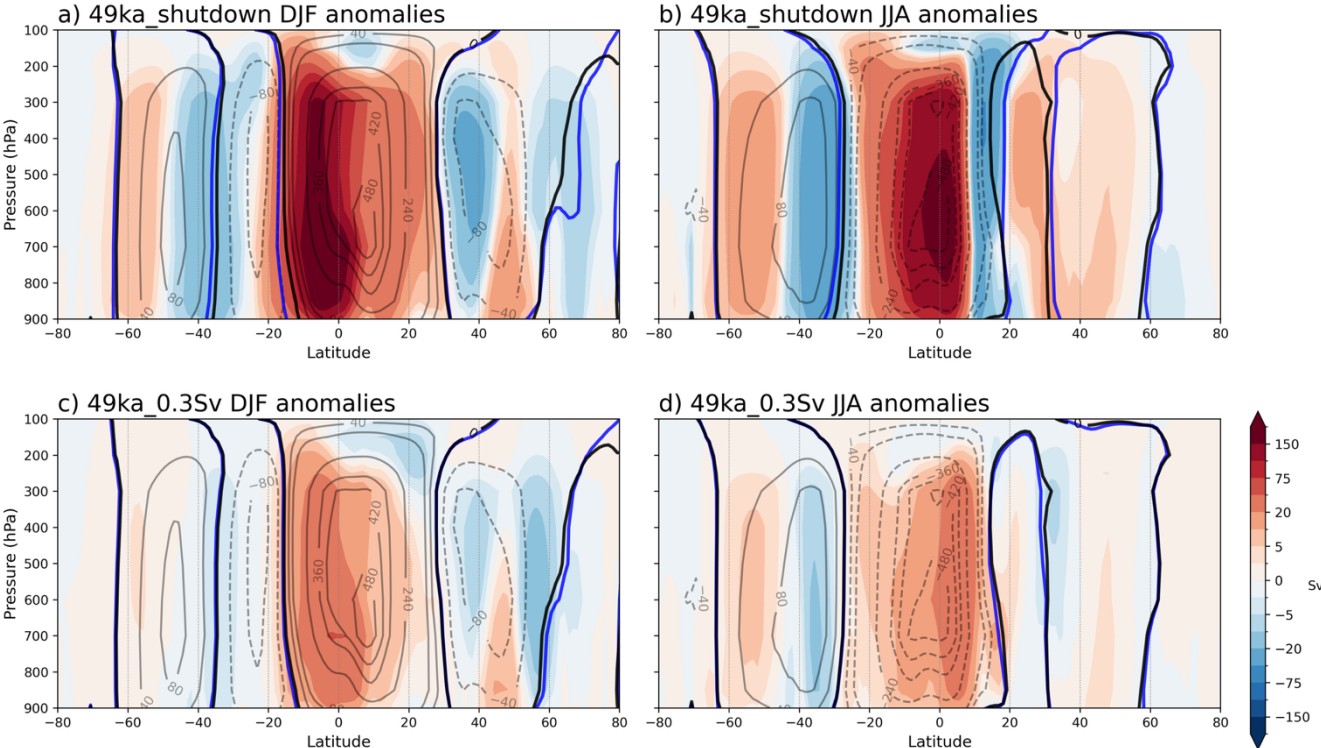



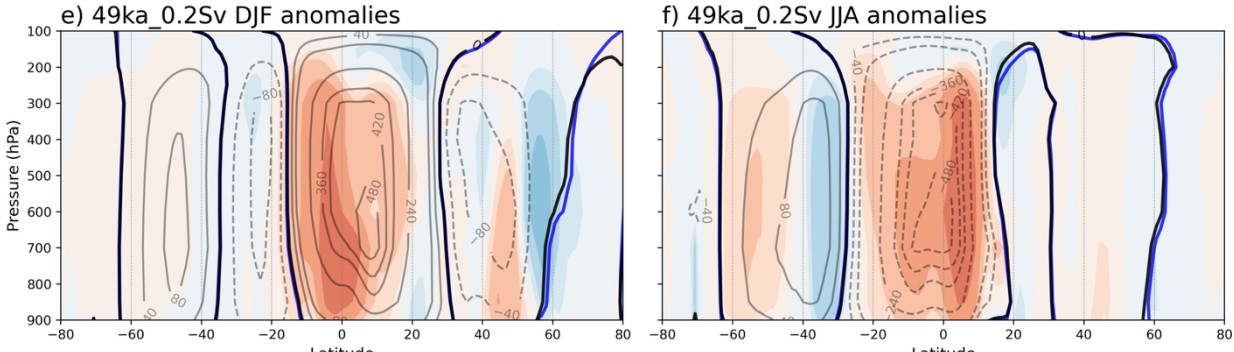

**Figure 4:** DJF and JJA 49ka_shutdown (a, b), 49ka_0.3Sv (c, d), and 49ka_0.2Sv (e, f) minus 49ka_control atmospheric
mass streamfunction anomalies (Sv, shading). Black contours (solid=positive; dashed=negative) are 49ka_control absolute
streamfunction values. The thick black and blue lines represent the zero contour in 49ka_control and each freshwater hosing
experiments, respectively.

In austral winter (JJA), the SH HC strength is reduced by 32.5 % in 49ka_shutdown, compared to 4 % and 2.8 % in
the 49ka_0.3 Sv and 49ka_0.2 Sv slowdown experiments, respectively (Fig. 5). The negative anomalies simulated in
49ka_shutdown experiment along the ascending branch of the SH HC around 18.4° N (Fig. 4b) are not seen in the slowdown
simulations (Fig. 4d & 4f). This may be attributed by a 3.5° northward shift of the ascending branch of the SH HC due to
AMOC shutdown, leading to an expansion of the SH HC width by 11.7 %. A 2.9 % reduction in the SH HC meridional
extent is simulated in 49ka_0.3Sv due to southward shift in the SH HC ascending branch, while no change is observed in the
49ka_0.2 Sv experiment results (Fig. 5). All these changes suggest a non-linear response in the HC between the slowdown
and shutdown simulations.





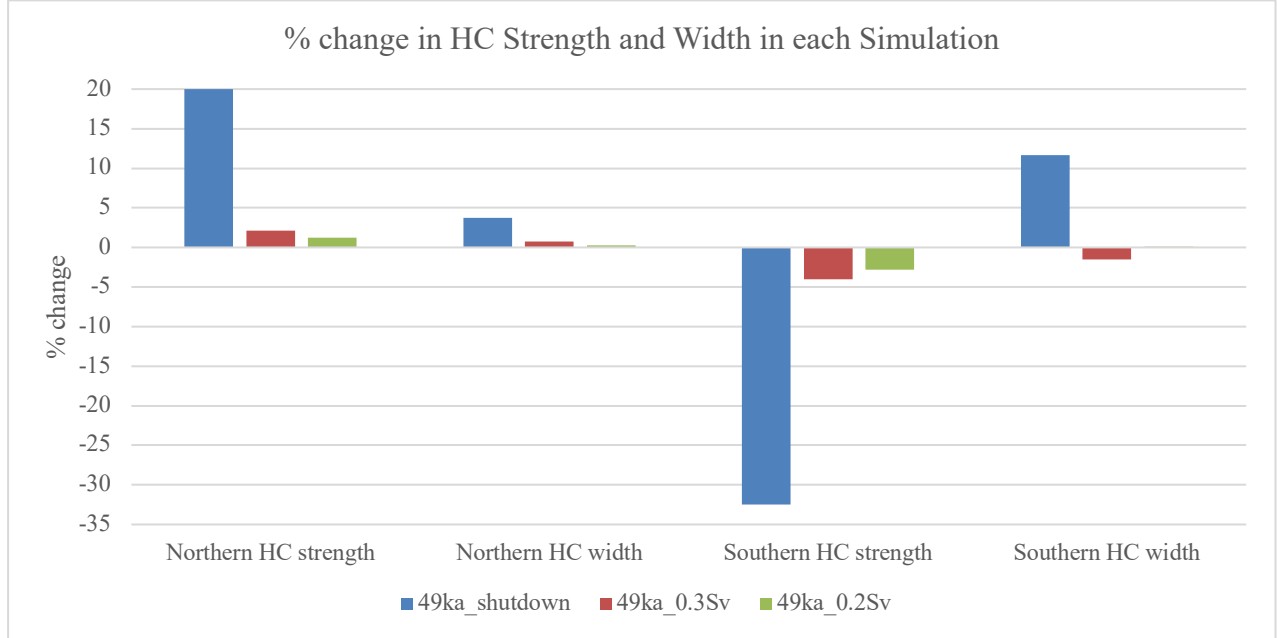

**Figure 5:** Percentage of changes in NH wintertime (DJF) HC and SH winter (JJA) HC strength and width in
49ka_shutdown, 49ka_0.3Sv and 49ka_0.2Sv simulations at 500 hPa (detailed values are in Table S2). Method for the
calculations is described in Section 2.4.

### 3.2.2 Changes in the subtropical pressure systems and circulation

Changes in the HC strength and width influence changes in the STR and surface wind stress associated with the subtropical
descending branch of the HC. In the shutdown simulation, the position of the southern JJA HC descending branch is shifted
southwards by 1.5° from 27.2° S with reduced strength while the northern DJF HC strength is enhanced with a small
northward shift (0.1°) in the position of the subtropical descending branch (Fig. 5 and Table S2). The position of the
descending branches in the slowdown simulations changes less than 0.3° (Table S2).

In the austral winter (JJA) season, the SH MSLP at ~27° S weakens in all simulations in response to a weaker
southern HC (Fig. 6b, 6d, 6f), with some slight strengthening over the Indian Ocean in the slowdown simulations (Fig. 6d,
6f). The upper-level subtropical jet at 200 hPa weakens as a response of weakened southern HC (Fig. S4b, 4d, 4f). The
subpolar low pressure belt shows negative surface pressure anomalies, while positive surface pressure anomalies of up to 5.7
hPa are found between 50° S and 70° S in the Indian and Pacific basins (Fig. 6b). In the NH, a region of low surface pressure
with negative MSLP anomalies of up to 7.2 hPa in the North Pacific subtropics is simulated in the 49ka_shutdown
experiment in JJA (Fig. 6b), associated with significantly increased precipitation (Fig. S5f), which corresponds to a
northward displacement of JJA ITCZ position to ~20.1° N. The surface winds adjust to changes in the surface pressure and





ITCZ shifts, generating an anomalous subtropical anticyclonic circulation over the North Pacific subtropical regions in the shutdown experiment (Fig. 6b), which is less evident in the slowdown simulations due to smaller changes in the southern HC (Fig. 6d, 6f).

In DJF season, there is an increase in pressure at northern subtropical regions in response to an enhancement of the descending branch of the northern HC in all simulations (Fig. 6a, 6c, 6e). The upper-level subtropical jet strengthens in the NH across all experiments (Fig. S4a, c, e), associated with stronger subtropical cyclonic circulation and positive surface pressure anomalies over Europe and North Atlantic (Fig. 6a, 6c, 6e). A decrease in surface pressure in the SH subtropical regions is simulated in all simulations, with less clear changes over the Indian Ocean in the 49ka_0.2Sv experiment due to

smaller southward shifts in DJF ITCZ positions at ~13° S. In the 49ka_shutdown experiment, it is evident that the surface wind changes are consistent with the southward shift in the ITCZ to ~15.8° S, particularly over the Indian and South Atlantic Oceans, with westerly anomalies to the north while easterly anomalies to the south, which generates anomalous cyclonic circulation over the Indian and south Atlantic Oceans in 49ka_shutdown experiment, associated with the weakening of surface pressure (Fig. 6a). In comparison to 49ka_shutdown, the changes in MSLP patterns and circulation between the

49ka_0.2Sv and 49ka_0.3Sv simulations show relatively linear relationships to AMOC weakening (Fig. 6c, 6e and 6d, 6f).

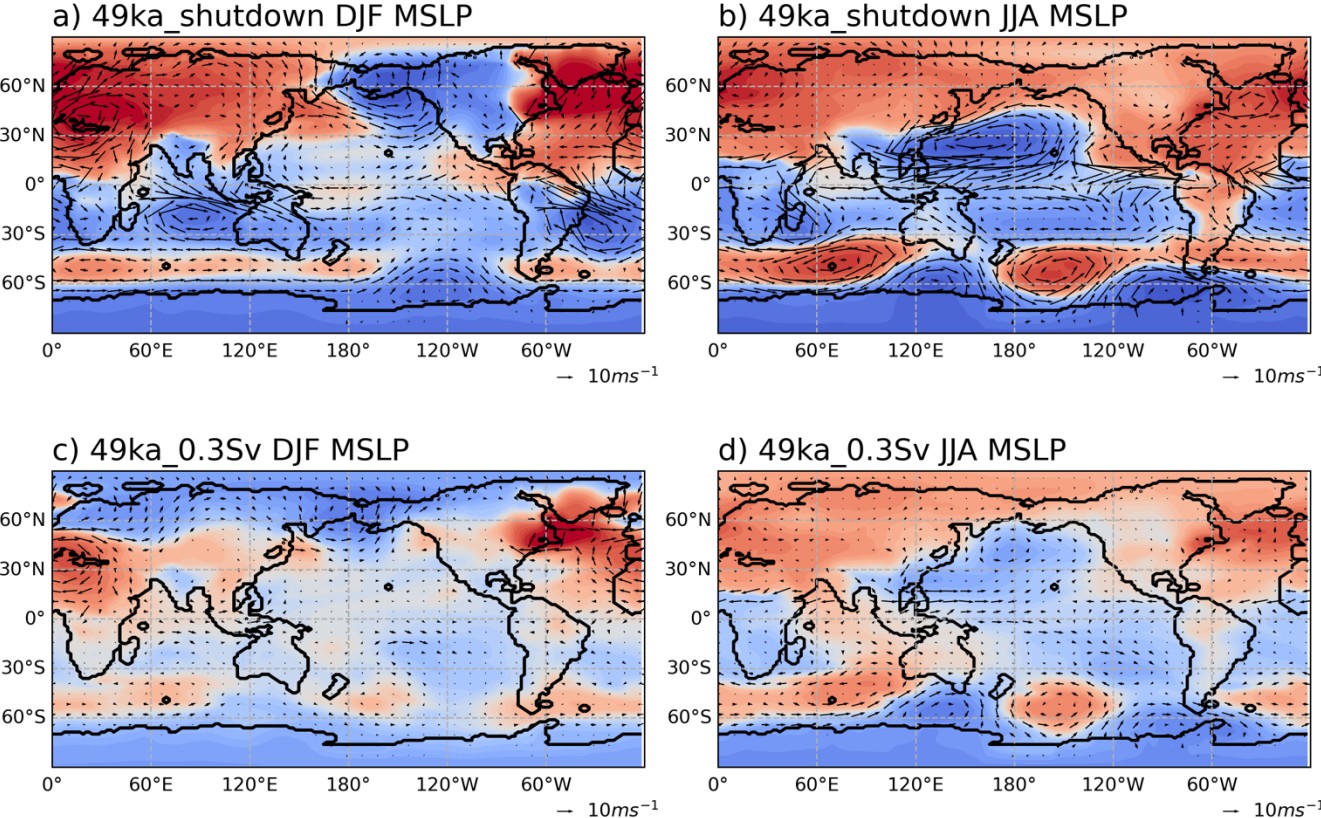



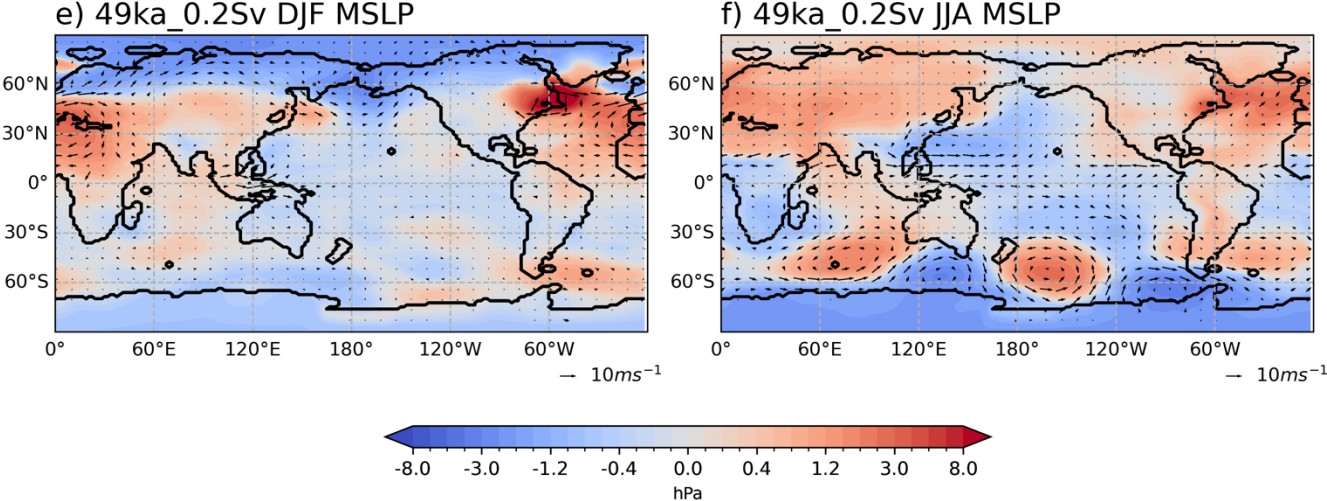

**Figure 6:** DJF (left column) and JJA (right column) MSLP anomalies (hPa) in each simulation relative to 49ka_control, with 850 hPa wind vector anomalies.

### 3.2.3 Changes in the SH mid-latitude westerly winds

In this section, we explore the impacts of the altered HC on SH mid-latitude westerly winds. Changes in the position and intensity of the SH westerly winds are evaluated, defining the SH westerlies strength as the zonal mean maximum wind speed between 40º S to 60º S at 850 hPa and the position of the SH westerly winds as the latitude of the maximum wind speed, interpolated to 0.5º latitude resolution.

The globally averaged SH westerlies strength increases by 11.4 % and shifts southwards by 6.5º to ~52º S in 49ka_shutdown simulation in JJA season, due to the weakening of the southern winter HC and southward displacement of its descending branch (Fig. 7a). Similarly, global mean strengthening and southward displacement of the SH westerlies are simulated in DJF season (Fig. 7e), but with smaller responses to AMOC weakening compared to JJA.

The westerlies strength shows the strongest intensification over the South Pacific Ocean in both seasons, with more than 21 % at around 55º S in JJA (Fig. 7c) and 6 % at ~51º S in DJF (Fig. 7g). There is little change in the meridional position over the Pacific basin (Fig. 7c, g), while the SH westerlies in JJA shift southwards by 6º and 5º over the Atlantic and Indian Oceans, respectively (Fig. 7b & 7d). Over the Atlantic basin, the westerlies strength reduces in the slowdown simulations in both seasons relative to 49ka_control but seems to only increase back to 49ka_control levels when the AMOC is shut down in JJA season (Fig. 7b), with relatively unchanged strength in response to AMOC shutdown in DJF (Fig. 7f). This is different from other ocean basins, which experience enhanced westerly strength in the slowdown simulations relative to 49ka_control.



In summary, the SH westerlies strength and meridional position in the AMOC slowdown experiments (49ka_0.2Sv and 49ka_0.3Sv) show little differences in their response to AMOC weakening (Fig. 7), despite some variations over the Atlantic and Indian Oceans in JJA due to slightly different changes in the southern winter HC. The strength is slightly strengthened by less than 6 % over the Pacific and Indian basins with no meridional displacement in the slowdown experiments relative to 49ka_control in JJA (Fig. 7c, d), while weaker and southward SH westerlies is simulated over the

Atlantic (Fig. 7b). Relatively no change is simulated in the slowdown simulations in DJF (Fig. 7e-h). However, when the AMOC is shut down, the SH westerlies response seems to follow the tendency over the Pacific Ocean with increased strength in both seasons (Fig. 7c, g), however, more complex responses are simulated in the other ocean basins.

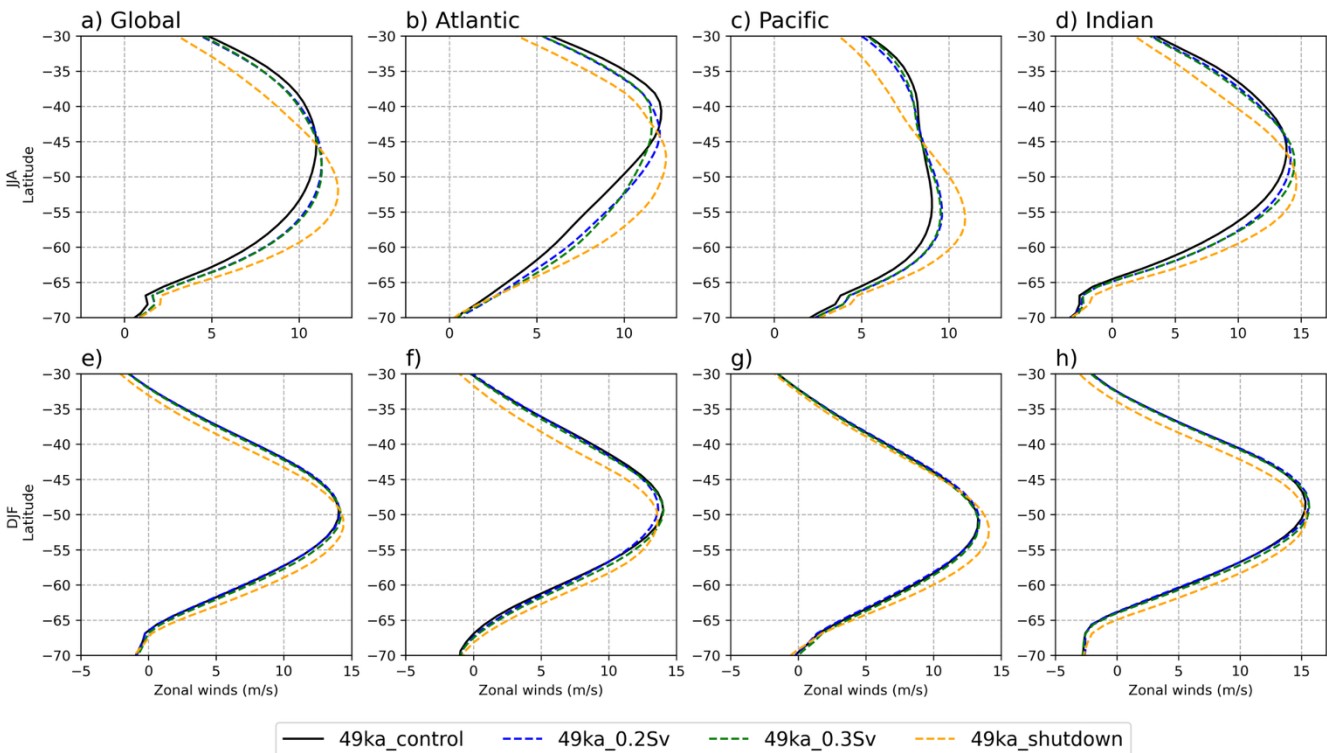

**Figure 7:** Southern Hemisphere JJA (a-d) and DJF (e-h) 850 hPa average westerly wind speed (m/s) in each simulation relative to 49ka_control (in black) globally, and across each ocean basin (longitude range defined in Section 2.4).

### 3.3 Impacts on seasonal Australasian climate

In this section, we investigate the climatic changes over Australia and the surrounding region in response to AMOC

slowdown and shutdown at 49 ka and propagated by large-scale changes in atmospheric circulation discussed above.




The northern half of the Australian continent, as well as parts of New Guinea, are simulated to be colder while the southern part of Australia is warmer in the shutdown simulation in both DJF and JJA seasons (Fig. 8a, 8d). This is consistent with previous AMOC weakening experiments performed with the same model under interglacial background climates (Du et al., 2025; Saini et al., 2025a). However, the amplitude of this temperature gradient is smaller in the slowdown experiments

(Fig. 8b-c, 8e-f). The 49ka_0.2Sv experiment displays a similar pattern as 49ka_shutdown in DJF, even though the changes are in-significant (Fig. 8c). In JJA, an overall cooling pattern is simulated across Australia in the slowdown experiments compared to significant warming anomalies are simulated over southern Australia due to AMOC shutdown (Fig. 8d-f).

Over New Zealand, significant warming is triggered by AMOC shutdown (Fig. 8a, 8d), while small temperature changes are suggested in the slowdown experiments, with significant cooling to the south and warming to the east of New

Zealand (Fig. 8b-c, 8e-f).

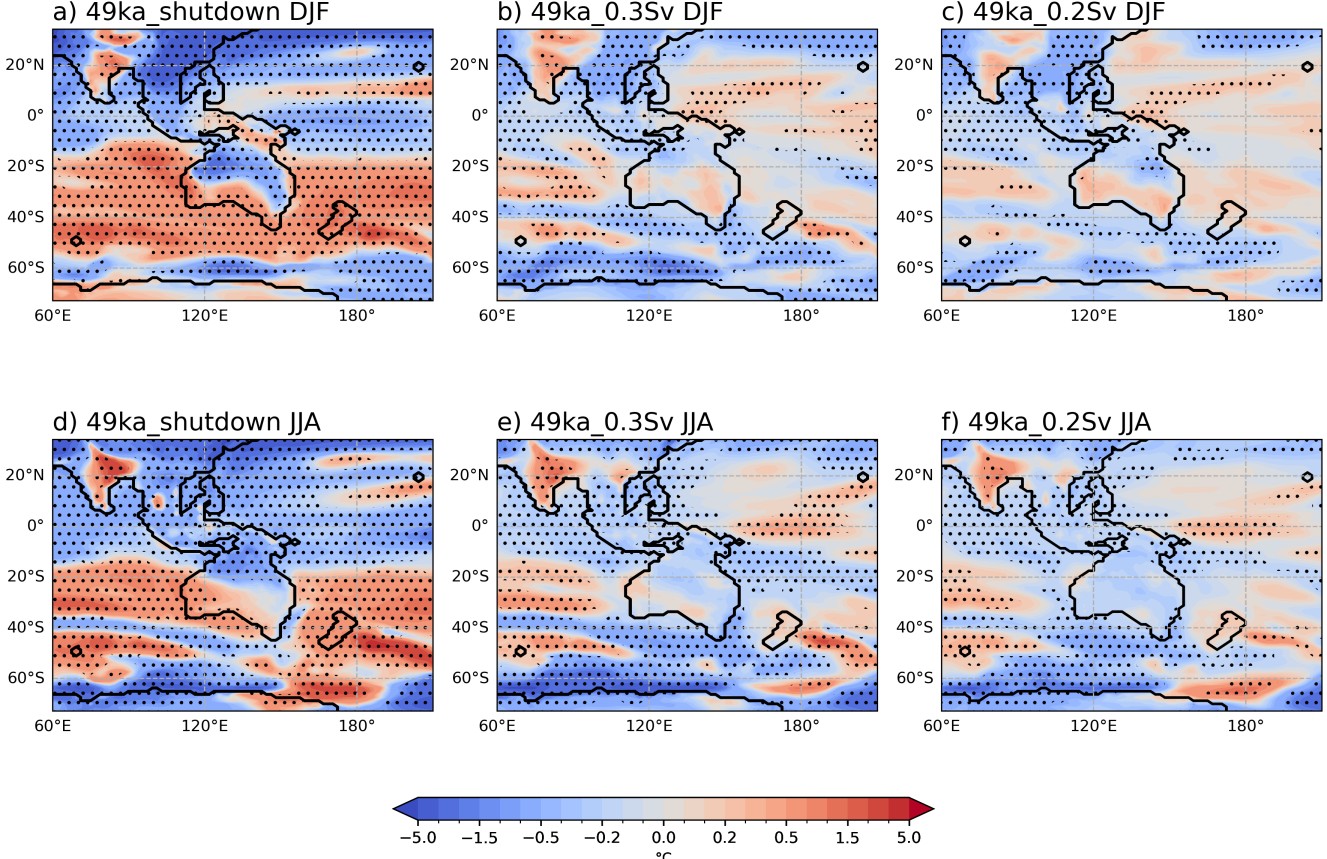

**Figure 8:** Australian annual (left column), DJF (middle) and JJA (right column) surface air temperature (SAT; in °C) anomalies in each simulation relative to 49ka_control. Stippling indicates areas with significant changes at the 95 %

confidence level according to a Student's t-test.





The current hydroclimate of northern Australia and nearby Maritime Continent and New Guinea are influenced by the Indo-Australian summer monsoon in the austral summer (DJF), while southern Australian and New Zealand hydroclimate is more affected by a northward shift of the SH westerly winds in austral winter (JJA) season. Moreover, the

STR, as the surface descending branch of the Hadley Cell, undergoes meridional displacement with the seasons, greatly influencing the timing of moisture delivery over Australia (Grose et al., 2015; Pepler et al., 2018).

In the shutdown experiment, significantly increased precipitation in DJF is simulated across the Australian continent south of 10 ºS, with an up to 6 mm/day increase in northern Australia due to AMOC shutdown (Fig. 9a). The increase is likely driven by stronger north-westerly winds associated with an enhanced and expanded Indo-Australian

summer monsoon (see Fig. 9a, where blue contours represent 49ka_control monsoon domain, and red contours represent the monsoon domains in the shutdown and slowdown experiments). The STR shifts southwards by 1º to around 41º S, allowing more tropical moisture to reach central Australia while drier conditions prevail around the high pressure regions (Fig. 9a). The STR shift is associated with significant cooler summer (DJF) temperatures of more than 2 ºC over northern Australia and New Guinea due to increased precipitation and cloud, whereas most of the SH mid-latitudes are warmer (Fig. 8a).

In the slowdown simulations, parts of northern Australia and New Guinea receive increased monsoon precipitation (Fig. 9b, 9c), but the monsoon spatial domains show little change relative to the 49ka_control. No significant ITCZ shift is simulated over the western Pacific, thus no significant changes in precipitation are simulated over New Guinea.

During the austral winter (JJA), significant drying is simulated along the east coast of Australia, extending to central Australia for all simulations (Fig. 9d-f). This drying trend is due to slightly increased pressure across Australia and New

Zealand (Fig. 6b) and a 1º southward displacement of the STR to ~30º S, associated with a displaced descending branch of the southern HC in JJA. The southward shift of the STR suppresses the mid-latitude frontal systems associated with low pressure that brings the cold south-westerly winds and precipitation into southern Australia in winter, as occurs under the modern climate. These changes in the STR also lead to drying conditions in southeastern Australia in the shutdown simulation (Fig. 9d). There is a consistent drying pattern over southern Australia in JJA across all experiments, which may

be due to the southward shift of the SH westerlies in all simulations (Fig. 7a). The STR position and intensity remains unchanged in the slowdown simulations in both seasons.



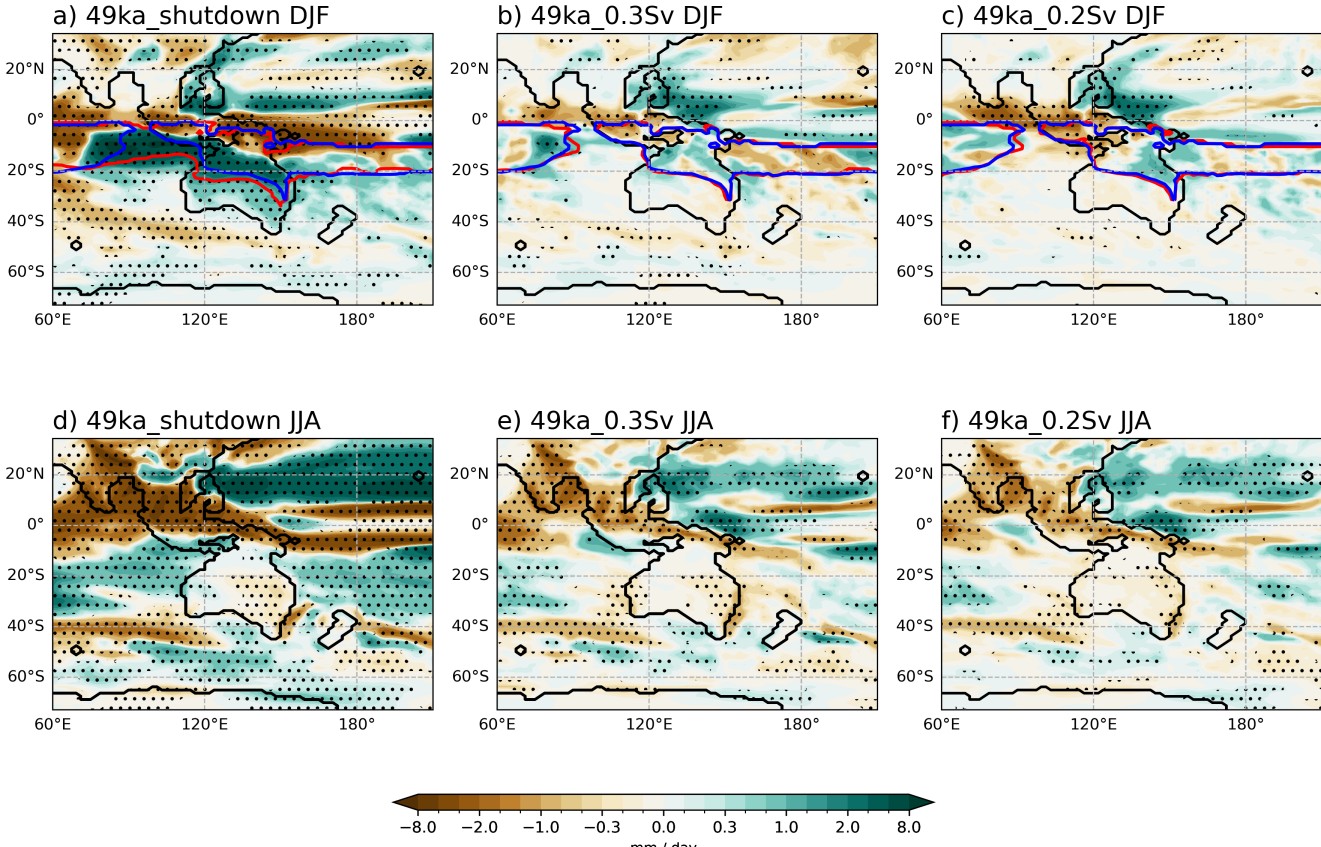

**Figure 9:** Australian (left column), DJF (middle) and JJA (right column) precipitation anomalies in each simulation relative to 49ka_control (mm/day). The blue contours in DJF (a-c) represent the 49ka_control monsoon domains while the red contours indicate monsoon domains in each hosing experiments Stippling indicates areas with significant changes at the 95 % confidence level according to a Student's t-test.

## 4 Discussion

In this study, we performed freshwater hosing experiments to slow down and shut down the AMOC under glacial conditions at 49 ka to represent D-O and HSs, respectively. We explored how the large-scale climate features, such as the atmospheric circulation and precipitation respond to different magnitudes of AMOC weakening and how it leads to changes in SH regional climates.

### 4.1 Climatic response to AMOC shutdown in HS simulation

In response to an AMOC shutdown at 49 ka, the annual mean position of the ITCZ moves southwards due to a strong interhemispheric temperature gradient, with cooler NH and warmer SH. The winter (DJF) northern HC strengthens and





increases in width, with a southward shift in its ascending branch (ITCZ position), while the southern wintertime (JJA) HC weakens but increases in width due to a small northward shift in JJA ITCZ position. Changes in the subtropical surface pressure are associated with variations in the descending branch of the HC. An increase in DJF MSLP is simulated in the NH subtropics due to enhanced descending branch of the northern HC, while the opposite weakening of the STR is seen in the SH (Fig. 10). A strong decrease in surface pressure accompanied by anomalous anti-cyclonic circulation is simulated over North Pacific in JJA, associated with significantly increased precipitation over the region (Fig. S5f). The upper-level SH subtropical jet weakens in JJA as a result of weakened southern winter HC. Global mean SH mid-latitude westerly winds at 850 hPa strengthens and migrating southwards in both JJA and DJF seasons, which further leads to strengthened upwelling of the cold deep waters across the Antarctic circumpolar latitudes and drives surface Southern Ocean cooling (Anderson et al., 2009).

The STR over the Australian region is weaker and moves southwards in both seasons. The austral summer (DJF) STR sits south of the Australian continent during the HS (Fig. 10a), which allows tropical moisture from expanded Indo-Australian monsoon domain to reach southern Australia, leading to significantly enhanced precipitation across the Australian continent south of ~10º S. A southward shift in the winter (JJA) STR reduces the fronts that pass over southern Australia, leading to drier conditions over southern Australia along with the enhanced and southwards displacement of the SH westerlies (Fig. 10b).

Our findings of enhanced northern HC and weaker southern HC in response to AMOC weakening are consistent with previous modelling studies (Chiang et al., 2014; Chiang and Friedman, 2012; Lee et al., 2011; Saini et al., 2025a), however, the simulated width of the southern HC due to an AMOC shutdown is expanded under 49 ka rather than reduced under Last Interglacial and PI boundary conditions in the same model (Saini et al., 2025a). This difference may be due to the different background states between warm interglacial and cold glacial climates, with more investigation needed in the future. Nevertheless, the simulated atmospheric circulation response to AMOC shutdown agrees with previous modelling studies, suggesting weaker South Pacific jet, stronger SH westerlies (Bard and Rickaby, 2009; Chiang et al., 2014; Lee et al., 2011), and increased SH monsoon precipitation (Cheng et al., 2012; Zhang et al., 2023) due to AMOC weakening.



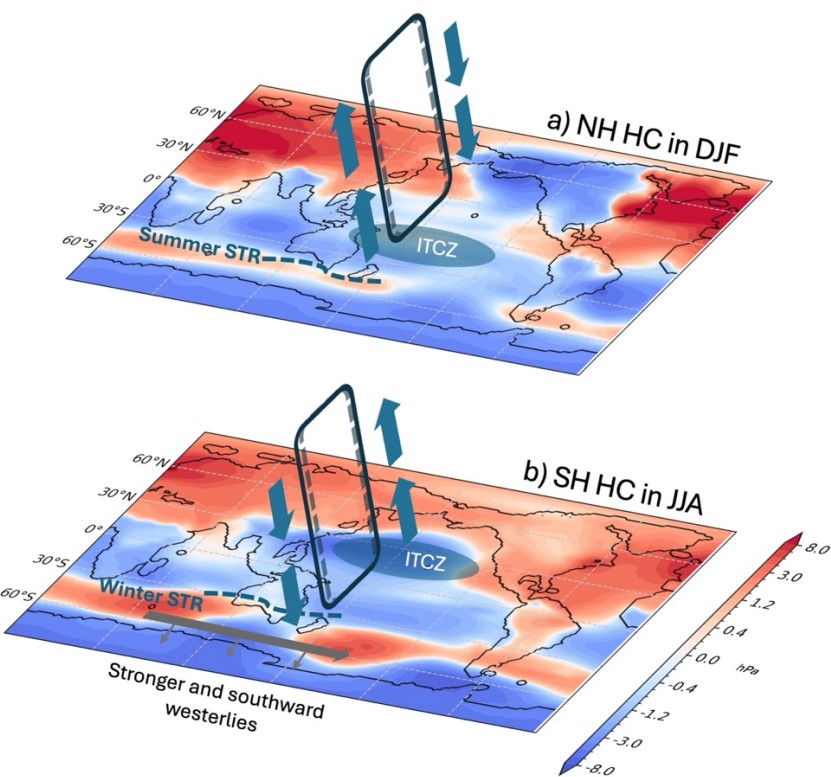

**Figure 10:** Schematic of changes in (a) NH HC in DJF, and (b) SH HC in JJA in the 49ka_shutdown simulation, plotted over MSLP (hPa) anomalies relative to 49ka_control. Dashed lines represent 49ka_control, solid lines represent 49ka_shutdown HC.

## 4.2 Non-linear climatic response between D-O and HSs

Global temperature and precipitation anomalies display non-linear patterns between D-O and HSs (as seen in Fig. 3). The Australian temperature field between the slowdown and shutdown experiments shows relatively linear changes (Fig. 8), but this is not the case for hydroclimates, particularly during the DJF monsoon season (Fig. 9). This may be associated with a different response of regional atmospheric systems to AMOC changes, such as altered Hadley circulation strength and differing interhemispheric temperature gradients and energy transport between the shutdown and slowdown simulations. Future studies performing longer simulations would be useful to explore slower responses, such as sea ice extent and Southern Ocean temperatures. Comparison of simulations from different climate models would also provide evidence of the robustness of model responses to AMOC weakening under glacial conditions.

We use normalised temperature and precipitation response to the AMOC decrease to quantify the linearity of climatic changes. The global area-averaged precipitation and temperature anomalies show linear changes per Sv of AMOC





decrease in AMOC slowdown simulations at 49 ka, with roughly -0.003 mm/day and -0.05 °C per Sv change for both the 0.2
Sv and 0.3 Sv experiments, respectively (see more details in Table S1). However, crossing the threshold of AMOC shutdown
between 0.3 Sv and 0.4 Sv triggers a different response in precipitation and temperature patterns which do not continue this
linear response either globally or regionally (Fig. 2a, 2d). The magnitude of global annual changes in precipitation and
temperature in the shutdown experiment are 1.3 times as large per Sv relative to the slowdown simulations (Fig. 3 for spatial
patterns), with larger changes in the SH (Table S1). This appears to be due to the non-linear responses in the large-scale
atmospheric circulations such as the Hadley Cell, leading to different responses in the climatic processes (e.g. pressure
system, westerly winds, etc.) between AMOC slowdown and AMOC shutdown. It is important to note that the location of
the threshold between linear weakening and nonlinear shutdown is likely to differ between models and climate states both in
the real world and in simulations.

### 4.3 Model-data comparison for D-O and HSs

We now compare our model simulations with available proxy records for D-O and HSs during MIS3 (Menviel et al., 2020),
focusing on temperature and precipitation anomalies in 49ka_shutdown versus the 49ka_0.3Sv experiment (as representative
of HS and D-O stadial respectively in Fig. 10). Our model simulation is overall consistent with these proxy records, as well
as with the proxy benchmarks as summarized by Izumi et al. (2023) in their Table 1 and Table 3, with colder and drier
conditions over the NH, weakened NH summer monsoon, and stronger South American and South African monsoons.
However, the monsoon responses from our simulations in the South American and South African monsoons domains show
more complex response (Fig. S6). Both D-O and HS simulations (Fig. 11c, 11d) are consistent with high-resolution proxies
of surface moisture from northern Queensland, Australia, suggesting wetter conditions during D-O and HSs (Denniston et
al., 2013, 2017; Treble et al., 2017). Moreover, our simulated  monsoon changes for HS (Fig. S6a) is consistent with proxy
records indicating weaker NH summer monsoon, and East Asian summer monsoon during HS5 (Chiang and Friedman,
2012; Dong et al., 2018). The simulated Indo-Australian summer monsoon is significantly enhanced in HS experiment (Fig.
S6b), which is consistent with a new speleothem record from southern Australia suggesting increased monsoon rainfall
during HS5 (Gould-Whaley et al., 2025, in prep.), and appears consistent with lacustrine record from Girraween Lagoon in
northern Australia (Bird et al., 2025).

Changes in the position and intensity of SH westerly winds are broadly consistent with results from previous
studies, with intensification of the SH westerlies in austral winter, being strongest over the South Pacific sector (Fig. 7c),
which is associated with a reduction in angular momentum transport from the tropics to the subtropics (Chiang and
Friedman, 2012). This strengthening of the SH westerly winds may explain the increase in atmospheric $CO_2$ concentration
during Heinrich 5 inferred from proxy records via the Southern Ocean (Menviel et al., 2014a; Wendt et al., 2024). In future
work, we plan to compare our simulation of Heinrich and D-O stadials with HS5 records in more detail.




**Figure 11:** Model simulations of annual surface air temperature (a, b; ºC) and precipitation (c, d; mm/day) anomalies in 49ka_shutdown (Heinrich stadial; left column) and 49ka_0.3Sv (D-O stadial; right column) simulations relative to 49ka_control with proxy records indicated by stars taken from Menviel et al. (2020).

## 5 Conclusions

This study investigates the climate response to AMOC slowdown and AMOC shutdown under 49 ka glacial conditions, corresponding to the time of HS5. The linearity of the temperature and precipitation responses to the magnitude of AMOC decrease is explored. There is a strong bipolar seesaw response due to AMOC shutdown with an average 3.8 °C cooling over the NH in HS. A southward shift of the annual mean ITCZ position is simulated, which leads to a stronger northern HC and weakened southern HC, influencing the climate response in the tropical and subtropical regions. Significantly increased precipitation is simulated over the entire Australian continent in summer (DJF) due to enhancement and southward expansion of the Indo-Australian summer monsoon and a southward shift of the STR, which allows tropical moisture to

reach southern Australia in HS. However, this is not seen in the AMOC slowdown simulations representing D-O stadials,
with insignificant increase in precipitation in northern Australia and drier conditions to the south. This is due to the subdued
response of HC strength and width. The JJA period is much drier over Australia in the HS simulation, possibly due to an
intensification and southward migration of the SH westerly winds. This drying pattern is also observed in the D-O stadial
simulations. The simulations are generally consistent with proxy records.

Our results suggest that the climate system responds linearly to a weakening of the AMOC, but once the threshold
of AMOC shutdown is crossed, a more complex atmospheric circulation and climate response is simulated. This study
provides a modelled climate response to AMOC shutdown under glacial conditions, enabling future comparisons under
alternative background climate states, and more detailed model-data investigations. Additional simulations of AMOC
weakening and shutdown under glacial conditions with other climate models would also provide a valuable comparison with
our results.

**Data availability**

All the model outputs used in this manuscript will be published on the DRYAD repository under the corresponding author's
account (https://datadryad.org/search?utf8=%E2%9C%93&q=Yanxuan+Du) and made publicly available upon acceptance
of the manuscript.

**Author contribution**

YD, JRB, LM, and RND designed the study. YD performed the modelling simulations (model configuration from HS and
DKH), conducted the analysis and wrote the manuscript with the input from JRB, LM, RND, DKH, and CNG provided
comments on the manuscript.

**Competing interests**

At least one of the (co-)authors is a member of the editorial board of Climate of the Past.

**Acknowledgements**

Yanxuan Du, Josephine R. Brown, Laurie Menviel, Himadri Saini, Russell N. Drysdale, and Calla N. Gould-Whaley
acknowledge the funding from the Australian Research Council (ARC) Grant DP220102134. Josephine R. Brown received
support from ARC Centre of Excellence for Weather of the 21$^{st}$ Century (CE230100012). Laurie Menviel acknowledges
support from ARC grant SR200100008. David K. Hutchinson acknowledges support from ARC grant DE220100279. This
research was supported by the Australian Government's National Collaborative Research Infrastructure Strategy (NCRIS),



with access to computational resources provided by the National Computational Infrastructure (NCI) through the National Computational Merit Allocation Scheme as well as the UNSW allocation scheme 10.26190/PMN5-7J50. This research was supported by the Research Computing Services NCI Access scheme at The University of Melbourne. The authors thank CSIRO for developing the ACCESS-ESM1.5 model configuration and making it freely available to researchers. This
research used the ACCESS-ESM1.5 model infrastructure provided by ACCESS-NRI, which is enabled by the Australian Government's National Collaborative Research Infrastructure Strategy (NCRIS).

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
