# Peer review of "Non-linear Climatic Response to the Weakening of the Atlantic Meridional Overturning Circulation During Glacial Times"

_EGUsphere, 2025_

## Author Comment (AC1)

**Response to Reviewer 1, Dr Shih-Yu Lee:**

*We thank Dr Shih-Yu Lee for her thoughtful and constructive comments to this manuscript.*

*Please see our point-by-point responses to each of the comments below in **blue and italic**, and suggested implementations in a revised manuscript in **green**. Please note that the line numbers refer to the original preprint and will be updated in the next iteration with the revised manuscript.*

**A) Experimental realism & robustness**

1.  Freshwater forcing magnitude, duration, and geometry: The shutdown uses 0.4 Sv for 500 yr over 50–70° N, 70–0° W (Table 2)—an idealized choice that likely exceeds plausible HE freshwater flux histories. Please (i) discuss physical plausibility versus "shock" idealization; (ii) provide a brief sensitivity or cite prior ACCESS ESM1.5 tests to hosing shape (pulsed vs ramped), duration, and release region (e.g., including/substituting the Nordic Seas or Labrador shelf). Even a short 100 yr/0.4 Sv pulse test or a reduced area hosing would help demonstrate that the non linear atmospheric reorganization at shutdown is not a by product of sustained extreme hosing.

*We thank the Reviewer for this comment and understand the concerns associated with the idealized experimental set up. We will add a new section in the Discussion to elaborate this concern.*

Line 486 (new Discussion section):

**4.4 Robustness of the simulations**

Our experimental setup was designed to assess the multi-centennial-scale impact of an AMOC shutdown and an AMOC slowdown on the climate system. The AMOC response to North Atlantic freshwater fluxes depends on both the magnitude and duration of the meltwater input. As shown in Du et al. (2025), a large, short North Atlantic freshwater pulse of 1 Sv for 5 years does not lead to significant AMOC changes, and the AMOC recovers quickly once the meltwater input is stopped. The experiments were designed to obtain a significant AMOC weakening and a shutdown with the smallest freshwater input. While this experimental setup is highly idealized and the input fluxes are significantly higher than current estimates (Zhou and McManus, 2024), it is similar to the experimental design of the recent North Atlantic Hosing Model Intercomparison Project (NaHOsMIP), in which 0.3 Sv is added in the North Atlantic for more than 100 model years to simulate the climate response to an AMOC shutdown in CMIP models (Ben-Yami et al., 2024; Diamond et al., 2025; Jackson et al., 2023).

Nevertheless, as seen in our experimental design, a pulse of 0.3 Sv for a few hundred years does not lead to an AMOC shutdown under 49ka boundary conditions. This highlights the issue of this model's sensitivity to freshwater forcing (Kageyama et al., 2013; Du et al. 2025; Saini et al., 2025a).

Given the previous experiments, with different North Atlantic meltwater input location, duration and magnitudes (Pontes & Menviel, 2024, Du et al, 2025, Saini et al., 2025), and the length of the simulations performed here (500 years), the two steps increase in meltwater input in 49ka_0.3Sv, from experiment 49ka_0.2Sv should not significantly affect the large-scale climate response, nor should the exact location of the meltwater input in the subpolar North Atlantic. A thorough study of the impact of meltwater input location on the climate response is out of scope of this study, but should be performed in the future.

2.  Internal variability and sampling: Results rely on 50 yr windows. Please consider a simple signal to noise check by resampling 50 yr blocks from the control and from each experiment (or show running 30 yr means across the last 150 yr) to demonstrate that key patterns (DJF ITCZ latitude, HC strengths/widths, STR latitude, SH westerly latitude) exceed internal variability.

*Thanks for the suggestion. We will add a paragraph in the new Discussion Section 4.4 (following the previous response) with a new supplementary Figure S8 to assess the statistical significance of the climate signals.*

Last paragraph in the Discussion Section 4.4:

Lastly, this study uses the last 50 years of each hosing experiment to assess the climate responses. When using the 30-year running-mean values for the last 150 years in each experiment, the results are consistent with our 50-year average (Fig. S8). We note that only one model was used, and a future study could examine the linearity in multiple models.

[Figure]

**Figure S8:** Box plot of 30-year means across the last 150 years for a) DJF ITCZ latitudes, and b) JJA SH westerly wind latitude in each experiment. Within each box, the thick line inside the box represents the median value (50[th] percentile) of each group; the top and bottom of the box shows the 25[th] and 75[th] percentile, respectively; the whiskers show 10[th] to 90[th] percentile.

**B) Mechanistic clarity/suggestion**

5.   Energetics of the ITCZ shift: Since the story hinges on interhemispheric energy transport compensation, would it be possible to exam TOA energy transport decomposition (DJF/JJA) that connects the ~1 PW reduction in NH ocean heat transport at 30° N to the southward DJF ITCZ jump in shutdown. Even a zonal mean cross equatorial energy flux figure would make the mechanism crisper.

*Thanks for the suggestion. We will add a new supplementary Figure S3 to illustrate the zonal-mean atmospheric heat transport to aid the discussion of the energetic mechanism of ITCZ shift in Section 3.1.*

Lines 210-213 (Section 3.1):

In response to an AMOC shutdown (Fig. 2d), a significant precipitation reduction is simulated to the north of the equator while the precipitation is greatly enhanced to the south, extending from the Atlantic to every ocean basin, indicating a southward shift of the mean ITCZ position. The atmospheric cross-equatorial heat transport increases (Fig. S3) to partially compensate for the reduced ocean heat transport (Fig. S3), pushing the ITCZ southwards towards the warmer hemisphere.

[Figure]

**Figure S3:** 49ka_shutdown - 49ka_control anomalies for annual mean, DJF, and JJA zonally averaged northward atmospheric heat transport (PW).

Lines 245-252 (Section 3.2.1):

In response to an AMOC weakening, the annual mean ITCZ position shifts southwards towards the warmer hemisphere in all simulations (Fig. 2d-2f), generating the cross-equatorial atmospheric energy transport (Hadley Cell) to partially compensate for the reduced oceanic heat transport. This drives alterations in the strength and width of the HC. Notably, the atmospheric compensation is highly seasonal. In the HS simulation, the TOA energy transport in DJF season reaches ~1.05 PW at 30 °N, while ~0.19 PW in JJA (Fig. S3) to compensate for the ~1 PW annual reduction in the North Atlantic ocean heat transport. This leads to more pronounced changes in DJF season in the HC and ITCZ responses.

Lines 490-491 (Conclusion):

Due to AMOC shutdown, a pronounced southward shift of the DJF ITCZ is simulated, associated with stronger DJF atmospheric compensation for the reduced NH ocean heat transport than in JJA. This asymmetric atmospheric response leads to a strengthened northern winter (DJF) HC and weakened southern wintertime (JJA) HC, influencing the seasonal climate response in the tropical and subtropical regions.

6. Southern Ocean/sea-ice feedbacks: You note notable SAT cooling near the Ross/Weddell sectors and stronger SH westerlies in shutdown (Fig. 2a; Fig. 7). The argument will be more comprehensive to show how wind/ice/ocean coupling amplifies the SH response.

*Thanks for the suggestion. We will add a statement in Section 3.2.3 to demonstrate the impact of the high southern latitude cooling on the westerlies.*

Line 326 (add to the end of the paragraph):

The strengthening of the westerlies is driven by an enhanced meridional temperature gradient in the Southern Ocean due to high-latitude cooling near the Ross/Weddell sectors (Fig. S4).

7. Basin contrasts in SH westerlies: The Atlantic sector behaves differently from the Pacific/Indian in slowdowns; shutdown realigns them (Fig. 7). Would be good to clarify why the Atlantic deviates.

*Thanks for the suggestion. This is due to the stronger and clearer temperature response in the Atlantic basin that leads to changes in the SH westerlies. Since the AMOC reductions in the slowdown experiments are small, the temperature response is limited to the Atlantic basin, which leads to larger changes in the SH westerlies in the Atlantic basin compared to the others where there is little change in the interhemispheric temperature gradient.*

*We will add more discussion in the corresponding Section 3.2.3.*

Line 334 (add to the end of the paragraph):

This may be due to the stronger interhemispheric temperature gradient change in the Atlantic Ocean in the slowdown simulations compared with the other ocean basins.

8. Carbon cycle configuration and outputs: ACCESS ESM1.5 includes an interactive carbon cycle, and the author discuss potential $CO_2$ links via SH westerlies (p. 22). Please state whether carbon was prognostic or prescribed here and, if active, showing simulated air–sea $CO_2$ flux (Southern Ocean sectors), DIC/alkalinity, and atmospheric $CO_2$ response will be of great interest.

*While for the radiative forcing, atmospheric $CO_2$ is kept constant at 284.3 ppm, the model also includes a prognostic atmospheric $CO_2$ tracer. However, this research question is the focus of other studies (e.g. Willeit et al., 2025), and it is out of scope of the topic of the manuscript. This question is explored by another PhD student in our group.*

**Reference:**
*Willeit, M., Ganopolski, A., Kaufhold, C., Dalmonech, D., Liu, B., and Ilyina, T.: Earth system response to Heinrich events explained by a bipolar convection seesaw, Nat. Geosci., 18, 1159–1166, https://doi.org/10.1038/s41561-025-01814-0, 2025.*

---

## Author Comment (AC2)

**Response to Reviewer 2, Dr Marlene Klockmann:**

Review of Du et al - Non-linear Climatic Response to the weakening of the Atlantic Meridional Overturning Circulation During Glacial Times

Du and colleagues compare the response of the climate system to weakened AMOC versus a collapsed AMOC, representative of Greenland Stadials (in relation to Dansgaard-Oeschger events) and Heinrich Stadials. They find that both temperature and precipitation change approximately linearly, as long as AMOC is decreasing linearly. As soon as the AMOC weakening crosses a threshold such that the AMOC collapses, also the climate system response becomes non-linear. The study initially analyses global fields, with a second focus on the Southern hemisphere/Australasia. In response to AMOC shutdown, both the Northern and Southern hemisphere winter Hadley cells (HC) increase in width, while the northern winter HC strengthens and the southern winter HC weakens. This leads to an expansion of the Indo-Australian summer monsoon and increased precipitation over most of the Australian continent.

The study is mostly well written and the figures well designed. Studies on the impact of AMOC weakening are of great general interest and relevance. I have a few major and a list of minor issues that should be addressed before publication of the study.

*We thank Dr Marlene Klockmann for all her thoughtful comments and valuable suggestions, which would greatly improve the quality of the manuscript.*

*Please see our point-by-point responses to each of the comments below in **blue and italic**, and suggested implementations in a revised manuscript in **green**. Please note that the line numbers refer to the original preprint and will be updated in the next iteration with the revised manuscript.*

**Major comments**

1. Framing

In the abstract and the conclusions, the study is framed in the context of a potential future AMOC shutdown/threshold. How transferable are the results of this study to future climate, given that the background climate (glacial vs global warming) are fundamentally different? This is briefly mentioned in l.424-429, but deserves a some more discussion.

*Thanks for the suggestion. We will expand the discussion in l.424-429 (last paragraph) in Section 4.1, and address the question in the Conclusion in the revised manuscript.*

Line 431 (add to the end of the paragraph):

The results of this study can be an example of the climate response to a strong AMOC reduction that can be compared with available proxy records to better understand the processes at play.

Lines 500-502 (Conclusion):

This study provides a modelled climate response to AMOC shutdown under glacial conditions. Although the background state is fundamentally different from future warming, the relatively good agreement between the model response and the proxy records provides better understanding of the climatic processes arising from a strong AMOC weakening, providing a basis for future comparisons under alternative background climate states, and more detailed model-data investigations.

2. Mechanistic link between AMOC weakening and Hadley cell changes

In parts, the manuscript remains very descriptive. I am missing some discussion of the mechanism how the AMOC changes lead to the described ITCZ and HC changes. Is it only the change in heat transport and the resulting change in temperature gradients?

*Thanks for the question. Yes, it is the changes in the cross-equatorial energy transport due to AMOC weakening that leads to changes in the ITCZ and HC in the atmospheric energy balance (Bischoff and Schneider, 2014; Kang et al., 2008; Lee et al., 2011; Pedro et al., 2018).*

*In response to AMOC weakening, we would expect an increase in the northward cross-equatorial atmospheric heat transport to compensate for the reduction in the northward ocean heat transport. This shifts the ITCZ southwards and alters the strength and width of the HC.*

*We will add more discussion about the mechanism discussion in the Introduction, Results Section 3.1, Discussion Section 3.2.1, and Conclusion with a new supplementary Figure S3 which shows the atmospheric heat transport.*

Lines 77-78 (Introduction):

This anomalous cross-equatorial atmospheric heat transport partially compensates for the reduced northward oceanic heat transport (Bischoff and Schneider, 2014; Kang et al., 2008; Lee et al., 2011; Pedro et al., 2018).

**References:**

Bischoff, T. and Schneider, T.: Energetic Constraints on the Position of the Intertropical Convergence Zone, J. Climate, 27, 4937–4951, https://doi.org/10.1175/JCLI-D-13-00650.1, 2014.

Kang, S. M., Held, I. M., Frierson, D. M. W., and Zhao, M.: The Response of the ITCZ to Extratropical Thermal Forcing: Idealized Slab-Ocean Experiments with a GCM, Journal of Climate, 21, 3521–3532, https://doi.org/10.1175/2007JCLI2146.1, 2008.

Lines 210-213 (Result Section 3.1):

In response to an AMOC shutdown (Fig. 2d), a significant precipitation reduction is simulated to the north of the equator while the precipitation is greatly enhanced to the south, extending from the Atlantic to every ocean basin, indicating a southward shift of the mean ITCZ position. The atmospheric cross-equatorial heat transport increases (Fig. S3) to partially compensate the reduced ocean heat transport (Fig. S3), which pushes the ITCZ southwards towards the warmer hemisphere.

[Figure]

**Figure S3:** 49ka_shutdown - 49ka_control anomalies for annual mean, DJF, and JJA zonally averaged northward atmospheric heat transport (PW).

Lines 245-252 (Discussion Section 3.2.1):

In response to an AMOC weakening, the annual mean ITCZ position shifts southwards towards the warmer hemisphere in all simulations (Fig. 2d-2f), generating the cross-equatorial atmospheric energy transport (Hadley Cell) to partially compensate for the reduced oceanic heat transport. This drives alterations in the strength and width of the HC. Notably, the atmospheric compensation is highly seasonal. In the HS simulation, the TOA energy transport in DJF season reaches ~1.05 PW at 30 °N, while ~0.19 PW in JJA (Fig. S3) to compensate for the ~1 PW reduction in the North Atlantic ocean heat transport. This leads to large seasonal variations in the HC and ITCZ response, with more pronounced changes in the DJF season.

As a result, the NH winter (DJF) HC strengthens while the SH winter JJA HC weakens in all simulations (Fig. 4 & Fig. 5). As the ascending branch of the HC, the ITCZ at 850 hPa in DJF season shifts southward by 0.82° and 0.34°, respectively in the 49ka_0.3Sv and 49ka_0.2Sv slowdown simulations, compared to a significant southward shift of 3.57° to ~15.8° S in the 49ka_shutdown simulation (thick lines in Fig. 4a, 4c, 4e). The global mean position of the JJA ITCZ shifts northwards by 0.97° to around 20.14° N in 49ka_shutdown experiment, which may be dominated by changes in the NH summer monsoon systems since the energy constraint is weak (~0.19 PW). The JJA ITCZ response in the slowdown simulations is not significant.

Lines 490-491 (Conclusion):

Due to AMOC shutdown, a pronounced southward shift of the DJF ITCZ is simulated, associated with stronger DJF atmospheric compensation for the reduced NH ocean heat transport than in JJA. This asymmetric atmospheric response leads to a strengthened northern winter (DJF) HC and weakened southern wintertime (JJA) HC, influencing the seasonal climate response in the tropical and subtropical regions.

3. The results of this study in the broader context of previous literature

In the introduction (l.68-73), the authors summarise the findings of previous studies on the atmospheric response to glacial AMOC weakening/shutdown. It would help in highlighting the relevance of this study, to emphasise more strongly the open questions that this study addresses and in the conclusions also to emphasise the new knowledge gained by this study. I understood that the results or this study are broadly consistent with previous literature based on both climate models and proxies. But what is the key new insight from this study?

*Thanks for the suggestion. We are now better explaining in the Introduction the gaps of previous studies, namely the lack of study of the impact of an AMOC weakening on the SH atmospheric circulation, and particularly the Hadley cells.*

Line 73 (add to the end of the paragraph):

Nevertheless, the impact of an AMOC weakening on the global-scale atmospheric circulation from the topics to the SH mid-latitudes via the Hadley Cells remains less certain.

*The linearity of the climate response to an AMOC decrease is another key new insight from this study (Lines 499-500). Moreover, the simulation results provide a plausible mechanism of the hydroclimate response around Australasia due to AMOC weakening through changes in atmospheric circulation, which can be used to compare with high-resolution proxy records around Australia in future studies. We will emphasise the new insights from this study in the Conclusion as well.*

Line 488 (add after the first sentence):

The simulation results are relatively consistent with proxy records for global temperature and precipitation changes, presenting a plausible insight into the SH/Australasian hydroclimate responses to AMOC weakening via changes in the seasonal HC and atmospheric circulation.

4. Setup of the hosing simulations

The setup seems not 100% consistent. Does it affect the results that the hosing is increased successively in the weaker hosing experiments? Would the 49ka_0.3Sv have the same state if it was started from 49ka_control? Or vice versa, would the response to 0.4Sv hosing be the same if you had continued to successively increase the hosing?

*Thanks for the questions. We will add a new Discussion Section 4.4 to address them.*

And yes, 49ka_0.3Sv and 49ka_shutdown have the same total integration time, but 49ka_0.2Sv does not. And also the integration time under the same hosing strength is different for the two DO-analogues vs the shutdown experiment. It probably does not have a big effect on the results, but it would be good to have this at least discussed.

*Thanks for the suggestion, this will also be added in the new Discussion section.*

Line 486 (new Discussion section):

**4.4 Robustness of the simulations**

Our experimental setup was designed to assess the multi-centennial-scale impact of an AMOC shutdown and an AMOC slowdown on the climate system. The AMOC response to North Atlantic freshwater fluxes depends on both the magnitude and duration of the meltwater input. As shown in Du et al. (2025), a large, short North Atlantic freshwater pulse of 1 Sv for 5 years does not lead to significant AMOC changes, and the AMOC recovers quickly once the meltwater input is stopped. The experiments were designed to obtain a significant AMOC weakening and a shutdown with the smallest freshwater input. While this experimental setup is highly idealized and the input fluxes are significantly higher than current estimates (Zhou and McManus, 2024), it is similar to the experimental design of the recent North Atlantic Hosing Model Intercomparison Project (NaHOsMIP), in which 0.3 Sv is added in the North Atlantic for more than 100 model years to simulate the climate response to an AMOC shutdown in CMIP models (Ben-Yami et al., 2024; Diamond et al., 2025; Jackson et al., 2023).

Nevertheless, as seen in our experimental design, a pulse of 0.3 Sv for a few hundred years does not lead to an AMOC shutdown under 49ka boundary conditions. This highlights the issue of this model's sensitivity to freshwater forcing (Kageyama et al., 2013; Du et al. 2025; Saini et al., 2025a).

Given the previous experiments, with different North Atlantic meltwater input location, duration and magnitudes (Pontes & Menviel, 2024, Du et al., 2025, Saini et al., 2025), and the length of the simulations performed here (500 years), the two steps increase in meltwater input in 49ka_0.3Sv, from experiment 49ka_0.2Sv should not significantly affect the large-scale climate response, nor should the exact location of the meltwater input in the subpolar North Atlantic. A thorough study of the impact of meltwater input location on the climate response is out of scope of this study, but should be performed in the future.

5. Significance/Robustness of very small changes in latitude

In many occasions, very small changes in latitude are reported (0.1° or even smaller). Given that the latitudes are interpolated to 0.5°, are changes smaller than 0.5° even significant/robust/detectable? Is not any change <0.5° below the accuracy of the spatial resolution?

*Thanks for the questions. We agree that changes below 0.5° can be interpreted as non-significant changes, and they are mainly coming from the 0.2Sv and 0.3Sv experiments. The reason why we included the values in the text is to compare with the larger shift due to AMOC shutdown to evaluate linearity of changes.*

*We will remove values of very small changes (≤ 0.1°) and instead indicate them as not statistically significant in the text. An additional paragraph in a new Discussion Section 4.4 will also be added to assess the statistical significance of some of the climate signals with relatively stronger response with a new supplementary Figure S8.*

Last paragraph in the new Discussion section 4.4:

Lastly, this study uses the last 50 years of each hosing experiment to assess the climate responses. When using the 30-year running-mean values for the last 150 years in each experiment, the results are consistent with our 50-year average (Fig. S8). We note that only one model was used, and a future study could examine the linearity in multiple models.

[Figure]

**Figure S8:** Box plot of 30-year mean across the last 150 years for a) DJF ITCZ latitudes, and b) JJA SH westerly wind latitude in each experiment. Within each box, the think line inside the box represents the median value (50th percentile) of each group; the top and bottom of the box shows the 25th and 75th percentile, respectively; the whiskers show 10th to 90th percentile.

6. Consistent way of referring to the simulations

Sometimes "AMOC weakening" refers to only the 0.2SV and 0.3Sv simulation, sometimes it seems to refer to all three simulations, including the shutdown. This makes it sometimes hard to clearly follow the argument. You could e.g. consistently refer to the 0.2SV and 0.3Sv simulations as "the DO simulations" and to the shutdown simulation as "the HS simulation" and then be very clear, whether your current result applies only to the DO or HS simulations or to both. Other clear nomenclatures are of course possible as well.

*Sorry about the confusion, we will make the naming consistent throughout the paper and add a statement in the Method section to define the terminology in the manuscript.*

Line 155:

The 49ka_0.2Sv and 49ka_0.3Sv experiments are hereafter referred to as D-O stadial and/or slowdown simulation/experiment, and the 49ka_shutdown experiments are also referred to as HS and/or shutdown simulation/experiment. AMOC weakening refers to all three experiments.

7. Absolute changes vs changes normalised by AMOC change

This is more a question that I had while reading:  would it make sense to express changes in other variables (HC strength, width, etc) also as a function of AMOC change?

*Thanks for the suggestion. The reason why we only show the normalised changes for temperature and precipitation is that we aim to focus more on the teleconnection between HC and other elements of atmospheric circulation in this study.*

*Moreover, as seen in Fig. 5 and 7, and the ITCZ changes in the 0.2 and 0.3 Sv are too small to be detected. The normalised changes would thus be even less robust. Therefore, we only included the normalised changes for temperature and precipitation.*

**Minor**

l.17: "global mean temperature and precipitation anomalies increase linearly" This is not reflected in the main text, where you emphasise non-linearity also for the global mean fields (factor of 1.3 for both variables).

*Sorry for the confusion, in this sentence we meant to express that the anomalies increase linearly in the slowdown simulations; however, crossing the threshold of AMOC shutdown leads to non-linear changes.*

*We will rephrase the sentence and add a definition in the Method to define the AMOC weakening simulations (as for comment 6).*

Line 17:

Global mean temperature and precipitation anomalies increase linearly with the slowdown of the AMOC; however, crossing the threshold of AMOC shutdown results in non-linear and more complex atmospheric circulation and climate responses.

l.55: HS5 is introduced quite abruptly. Any particular reason why you focus on HS5?

*Thanks for the comment. One main reason we focus on HS5 is that we have a new high-resolution speleothem record from southern Australia which shows a strong signal for*

*increased moisture during HS5 (Gould-Whaley et al., 2025, in prep.), which can be compared with our simulations results. However, since this work is still in preparation, we cannot provide many details in this manuscript. We will rephrase this paragraph to reflect this context in the Introduction.*

Lines 55-63:

Many D-O and HSs were documented during Marine Isotope Stage 3 (MIS3, ~59.4-27.8 ka; Sanchez Goñi and Harrison, 2010). Heinrich stadial 5 (HS5) occurred at a time (~48.8-47.6 ka; Menviel et al., 2014b; Sanchez Goñi and Harrison, 2010) during MIS3, received higher summer insolation in both hemispheres relative to pre-industrial (PI) levels due to greater obliquity (Berger, 1978). The atmospheric $CO_2$ concentration was lower (Köhler et al., 2017). Global sea level was ~60-65 m lower than PI (Shakun et al., 2015), with extensive Laurentide and Scandinavian ice sheets (Gowan et al., 2021). In northern Australia at Girraween Lagoon, paleoclimate records suggest relatively drier background climates during MIS3 relative to PI (Bird, 2025). This is generally consistent with model simulations of 49 ka (Saini et al., 2025b), indicating reduced precipitation to the north of northern Australia as a result of changes in all the boundary conditions (orbital, ice sheet albedo, etc.). Wetter HSs than interstadials are also suggested during MIS3 at Girraween Lagoon (Bird, 2025). This coincides with a new high-resolution speleothem record from southern Australia (Gould-Whaley et al., 2025, in prep.), suggesting increased moisture availability during HS5. However, it remains less clear how the climate responds to reduced AMOC at HS5 from model simulations and the consistency between simulation and proxy reconstructions for regional Australasian changes at HS5.

l.84-85: what do you mean by "consistent response here" what are the open questions?

*We will rephrase the sentence to:* "A consistent response in the strength and meridional shift of the SH westerlies to [...]".

Tab1: how are albedo, vegetation, topography and runoff obtained for 49ka? what are they based on?

*Sorry for the incompleteness, this section is now modified to clarify the 49 ka boundary condition changes.*

Lines 113-115:

The model was first integrated under 49 ka boundary conditions (Table 1, Saini et al., 2025b). The model is forced by 49 ka orbital parameters (Berger, 1978), greenhouse gas concentrations (Köhler et al., 2017), ice-sheet extent and topography corresponding to 52.5 ka (Gowan et al., 2021) with closest match to ~ 48-52 ka sea level estimates (Shakun et al., 2015). Vegetation was modified to reflect the associated continental ice cover and converting forest to bare soil between the Cordilleran and Laurentide ice-sheet and C3 crops were replaced by surrounding vegetation. The land-sea mask and river runoff are also adjusted to reflect the modified topography (see details in Saini et al., 2025b).

**Table 1.** Full boundary conditions for the 49 ka simulation relative to PI (reproduced from Saini et al., 2025b).

|  | 49 ka | PI |
|---|---|---|
| **Orbital parameters** |  |  |
| Eccentricity | 0.01292 | 0.01674 |
| Obliquity (°) | 24.435 | 23.459 |
| Perihelion – 180 (°) | 62.451 | 100.33 |
| **Greenhouse gases** |  |  |
| $CO_2$ | 199 | 284.3 |
| $N_2O$ | 237 | 273 |
| $CH_4$ | 432 | 284.3 |
| **Ice-sheet extent and salinity** | 49 ka | PI |
| **Albedo and vegetation** | 49 ka | PI |
| **Topography and runoff** | 49 ka | PI |

l.119-120: this sentence is not fully clear. The 1555 years are the spinup of the 49ka_control simulation? Or are the 760 years of 49ka_control part of the 1555 years? why were boundary conditions changed stepwise and how?

*Sorry about the confusion. We have changed the discussion here to clarify it. The reason why the boundary conditions were implemented step-wise is that in the previous study by Saini et al. (2025b), the authors compare the influence from different boundary conditions (e.g. NH ice sheets) on model's response in atmospheric circulation.*

Lines 119-122:

The 49 ka simulation was run with step-wise changes in the boundary conditions for a total of 1555 model years, with the last 760 years implemented with full boundary conditions (49ka-full in Saini et al., 2025b). The last 100 years in 49ka-full show relatively stable surface air temperature and sea-surface temperatures (Saini et al., 2025b, Fig. A1). We therefore consider this as a quasi-equilibrium state for 49 ka climate, and define this period as the 49ka_control experiment in this study.

l.113-122: This section relies heavily on information from Saini et al 2025b. While you of course don't need to go as much into the details as Saini et al, it shoud still be possible to understand the relevant aspects of the setup without having to read Saini et al in addition.

*Thanks for the suggestion, we have now modified this section as detailed in the previous two comments.*

l.140/Fig.S1: include 49ka_control in Figure S1 for comparison

*We have added it now.*

[Figure]

Figure S1: DJF mixed-layer depth (MLD; in m) in the North Atlantic for (a) 49ka_control, (b) the Heinrich stadial simulation (49ka_shutdown), and D-O stadial simulations (c) 49ka_0.3Sv, (d) 49ka_0.2Sv.

l.175: what is the additional benefit of the cubic spline? are the results very sensitive to the fitting parameters?

*This method is adopted from Grose et al. (2015), with the use of a fitted cubic spine, we would be able to be detect the location and pressure of maximum MSLP more precisely. We have checked the codes for computing the cubic spline, our results are not sensitive to the parameters chosen.*

l.188-189: was the AMOC reduction in the simulations with the 40% heat transport weakening of a similar magnitude as in 49ka_shutdown? If the residual AMOC strength after the "shutdown" was stronger than in 49ka_shutdown, this could be an explanation for the difference in heat transport reduction. Also, these numbers are probably very model dependent to begin with.

*Thanks for the suggestion, we will modify the discussion here.*

Lines 188-189:

The meridional oceanic heat transport to the North Atlantic is reduced by ~77 % (~1 PW at 30° N) in the HS experiment (see Fig. S2) This change is larger than previous model simulations of AMOC shutdowns under LGM conditions that found an average ~40 % reduction in Atlantic meridional heat transport (~0.8 PW at 30° N) (Menviel et al., 2008, 2020; Stouffer et al., 2006). The larger reduction in meridional oceanic heat transport can

be explained by the large AMOC reduction simulated here (29 Sv reduction compared to an averaged of 15 Sv in previous LGM experiments; Kageyama et al., 2013).

l.191: How is this huge change in the subpolar North Atlantic possible? is that sea-ice related? I can see that air temperatures over sea ice may change dramatically, but a similar cooling seems to be happening in the SST as well. This implies that the subpolar North Atlantic must have temperatures of around 25° or higher, is it really that warm? But perhaps the SST changes are not as large and only the non-linear spacing of the colourbar in Fig. S3 makes it hard to read the actual magnitude of the SST change.

*Sorry about the colorbar in Fig. S3, the maximum cooling in SST (-15.4 °C, Line 192) is indeed not as large as SAT. We have now changed it in Fig. S4.*

*We still prefer to use the non-linear spacing for the colorbar as it is easier to see the pattern of smaller temperature changes outside the high latitudes which may be important for precipitation and circulation changes in the tropics, e.g. ITCZ and monsoons.*

[Figure]

**Figure S4:** Annual mean sea surface temperature anomalies (SST; ºC) relative to 49ka_control in each simulation. Stippling indicates statically significant differences from the control at the 95 % confidence level according to the Student's t-test

*We have checked the location with the maximum SAT cooling of -26.3 °C in the 49ka_shutdown, at 55° N and 55° E. The absolute 49ka_control SAT at this location is simulated at 2.3 °C, while -24 °C at 49ka_shutdown. This can be seen in the warming anomalies at 49ka_control relative to PI (Figure 2 from Saini et al., 2025b) due to the higher AMOC strength. The sentence will be rephrased to reflect this.*

[Figure]

*Figure 2 in Saini et al. (2025b): Annual mean (a) SAT and (b) SST anomalies (°C) in 49ka_control relative to PI.*

Lines 190-192:

Reduced heat transport leads to a widespread significant NH mean cooling of 3.8 °C, with maximum annual mean surface air temperature (SAT) decrease of up to 26.3 °C in the Labrador Sea (Fig. 2a), while the SH display a significant warming (mean 0.43°C) from 0 to 55°S.

l.198: warming over Antarctica is also non-significant in 49ka_shutdown (no stippling over Antarctica).

*Thanks, we will rephrase the sentence.*

l.209: To me, the precipitation patterns do not seem more diverse than the temperature patterns. At least not on the global scale. Which is probably also reflected by the fact that the global mean temperature and precipitation both change by a factor of 1.3 between DO and HS simulations.

*Thanks, we will delete the sentence.*

l.257-259: this sentence should refer to the change in HC width, not the absolute HS width

*Thanks, this will be modified.*

l.300: this seems to be true mostly over Africa/the subtropical Atlantic/eastern pacific.

*Thanks, we will add the regions into the sentence.*

l.380-382: These two sentences seem to contradict each other "[…] Australia and New Guinea receive increased monsoon precipitation […]. […] thus no significant changes in precipitation are simulated over New Guinea."

*Thanks for pointing out, we will rephrase them.*

Lines 380-382:

*In the slowdown simulations, parts of northern Australia and New Guinea receive a small increase in monsoon precipitation (Fig. 9b, 9c), but the monsoon spatial domains show little change relative to the 49ka_control. No significant ITCZ shift is simulated over the western Pacific, thus no significant changes in precipitation are simulated.*

l.474-475: What were the previous studies based on?

*Thanks, we will add more details here.*

**Editorial**

*Thanks for the editorial suggestions, the below editorial comments will all be implemented in the revised manuscript.*

l.31-32 nomenclature: the sentence in its current form seems to imply that the warming/cooling transitions are the Greenland Interstadials/Stadials. But Interstadial/Stadial refers to the more or less stable periods between the transitions. Please reformulate

l.34 same as above. Also, stadial was already introduced in the sentence before.

l.40 "contain" instead of "contains"

l.40-43: please shortly explain what is debated and which usage you follow.

l.48: Add "The" before "climatic response" at the beginning of the sentence.

l.59: "global temperature was" instead of "is"

l.121-122: should this not be "relative stable surface air and sea surface temperatures" rather than "stable changes in SAT and SST changes"?

l.134: "more likely to close to a complete shutdown", something is missing in the sentence, please reformulate

l.197-199: sentence does not work. please reformulate

l.305-309: super long sentence. please break it down to smaller, easier sentences.

l.356-357: sentence does not work. please reformulate

l.383-391: difficult to follow which part applies to all or only the DO or HS simulations. Please reformulate for clarity

Fig.9 (caption): the identifiers "left column", "middle" and "right column" do not fit the structure of the figure. It should probably be only "top" and "bottom" behind DJF and JJA, respectively.

l.415: "strengthen and migrate" instead of "strengthens and migrating"